# Menopausal hormone therapy and the female brain: Leveraging neuroimaging and prescription registry data from the UK Biobank cohort

Claudia Barth[1]*, Liisa AM Galea[2,3], Emily G Jacobs[4], Bonnie H Lee[2], Lars T Westlye[5,6], Ann-Marie G de Lange[5,7,8]

[1]Division for Mental Health and Substance Abuse, Diakonhjemmet Hospital, Oslo, Norway; [2]Centre for Addiction and Mental Health, Toronto, Canada; [3]Department of Psychiatry, University of Toronto, Toronto, Canada; [4]Psychological and Brain Sciences, University of California Santa Barbara, Santa Barbara, United States; [5]Department of Psychology, University of Oslo, Oslo, Norway; [6]Centre for Precision Psychiatry, Division of Mental Health and Addiction, Oslo University Hospital, Oslo, Norway; [7]Department of Clinical Neurosciences, Lausanne University Hospital (CHUV) and University of Lausanne, Lausanne, Switzerland; [8]Department of Psychiatry, University of Oxford, Oxford, United Kingdom

*For correspondence:
claudia.barth@medisin.uio.no

Competing interest: The authors declare that no competing interests exist.

## eLife Assessment

This observational study from the UK Biobank provides an **important** investigation into the associations between menopausal hormone therapy and brain health in a large, population-based cohort of females in the UK. A **convincing** model of brain aging using an open source algorithm is used. While some modest adverse brain health characteristics were associated with current mHT use and older age at last use, the findings do not support a general neuroprotective effect of mHT nor severe adverse effects on the female brain. This work addresses a topic that is of grave importance since menopausal hormone therapy and its effect on the brain should be better understood in order to provide individualized effective medical support to women going through menopause.

## Abstract

**Background:** Menopausal hormone therapy (MHT) is generally thought to be neuroprotective, yet results have been inconsistent. Here, we present a comprehensive study of MHT use and brain characteristics in females from the UK Biobank.

**Methods:** 19,846 females with magnetic resonance imaging data were included. Detailed MHT prescription data from primary care records was available for 538. We tested for associations between the brain measures (i.e. gray/white matter brain age, hippocampal volumes, white matter hyperintensity volumes) and MHT user status, age at first and last use, duration of use, formulation, route of administration, dosage, type, and active ingredient. We further tested for the effects of a history of hysterectomy ± bilateral oophorectomy among MHT users and examined associations by APOE ε4 status.

**Results:** Current MHT users, not past users, showed older gray and white matter brain age, with a difference of up to 9 mo, and smaller hippocampal volumes compared to never-users. Longer duration of use and older age at last use post-menopause was associated with older gray and white

matter brain age, larger white matter hyperintensity volume, and smaller hippocampal volumes. MHT users with a history of hysterectomy ± bilateral oophorectomy showed *younger* gray matter brain age relative to MHT users without such history. We found no associations by APOE ε4 status and with other MHT variables.

**Conclusions:** Our results indicate that population-level associations between MHT use and female brain health might vary depending on duration of use and past surgical history.

**Funding:** The authors received funding from the Research Council of Norway (LTW: 223273, 249795, 273345, 298646, 300768), the South-Eastern Norway Regional Health Authority (CB: 2023037, 2022103; LTW: 2018076, 2019101), the European Research Council under the European Union's Horizon 2020 research and innovation program (LTW: 802998), the Swiss National Science Foundation (AMGdL: PZ00P3_193658), the Canadian Institutes for Health Research (LAMG: PJT-173554), the Treliving Family Chair in Women's Mental Health at the Centre for Addiction and Mental Health (LAMG), womenmind at the Centre for Addiction and Mental Health (LAMG, BHL), the Ann S. Bowers Women's Brain Health Initiative (EGJ), and the National Institutes of Health (EGJ: AG063843).

## Introduction

Ovarian hormones such as estrogens and progesterone fluctuate across the female lifespan with natural declines occurring at menopause, typically between the ages of 45 and 55. The cessation of ovarian function during the menopausal transition has been linked to an array of neural changes (*Brinton et al., 2015*), including a decline in brain glucose metabolism (*Ding et al., 2013*), reductions in gray matter (GM) and white matter (WM) volume (*Fjell et al., 2009*; *Goto et al., 2011*; *Mosconi et al., 2021*; *Mosconi et al., 2017*), and increased amyloid-beta deposition (*Mosconi et al., 2018*) as well as WM lesions (*Wen et al., 2009*). In combination with other risk factors, these neural changes might foster the emergence of neurodegenerative diseases such as late-onset Alzheimer's disease (AD), which is more often diagnosed in females relative to similarly aged males, with greater cognitive decline and neuropathological burden (*Laws et al., 2018*; *Laws et al., 2016*).

MHT is commonly prescribed to minimize vasomotor symptoms occurring during the menopausal transition and is generally thought to be neuroprotective, with a propensity to reduce the risk for AD (*Simpkins et al., 2009*; *Hogervorst et al., 2000*; *Zandi et al., 2002*; *Kim et al., 2021*) and improve cognition later in life (*Maki et al., 2011*). However, study results are equivocal (*Mills et al., 2023*), reporting both positive (*Schelbaum et al., 2021*; *Erickson et al., 2005*; *Ha et al., 2007*) and negative outcomes (*Kantarci et al., 2016*; *Lange et al., 2020*; *Pourhadi et al., 2023*). A 2021 study reported that reproductive history events signaling more estrogens exposure, including MHT use, were associated with greater gray matter volume in middle-aged females (*Schelbaum et al., 2021*), in line with other neuroimaging studies suggesting a protective effect of MHT on GM, WM, and ventricle size (*Erickson et al., 2005*; *Ha et al., 2007*). Conversely, MHT use has also been associated with greater atrophy (*Resnick et al., 2009*) and higher rates of ventricular expansion (*Kantarci et al., 2016*) in menopausal females. Similarly, in our previous UK Biobank study of ~16,000 females (*Lange et al., 2020*), we found positive associations between MHT use and older GM brain age, albeit with small effect sizes.

Besides mixed results in observational studies, randomized controlled trials (RCTs) such as the Women's Health Initiative Memory Study (WHIMS) suggest an increased risk of dementia and cognitive decline with MHT use. In detail, WHIMS found negative effects of prolonged oral administration of both conjugated equine estrogen (CEE) alone (*Shumaker et al., 2004*) or in combination with medroxyprogesterone acetate (MPA, synthetic progestin) (*Rapp et al., 2003*; *Shumaker et al., 2003*) among females aged 65 y or older. Similarly, the Heart and Estrogen/Progestin Replacement Study (HERS) showed an association between 4 y of CEE +MPA treatment and lower cognitive performance in older postmenopausal females (71±6 y) (*Grady et al., 2002*). In contrast, administering oral CEE or transdermal estradiol plus micronized progesterone in recently postmenopausal females did not alter cognition in the Kronos Early Estrogen Prevention Study (KEEPS) (*Gleason et al., 2015*). These mixed findings raise the question of whether a combination of timing, formulation, and route of administration might play a crucial role in the effectiveness of MHT.

**eLife digest** Ovarian hormones, including oestrogens and progesterone, fluctuate throughout each menstrual cycle, during and after pregnancy, and in the years leading up to menopause when ovarian function begins to decline. During these transitional years, up to 80% of women will experience symptoms such as hot flashes and night sweats, which are believed to stem from the brain.

Menopausal hormone therapy (MHT) often contains low doses of estrogens with or without progesterone and is commonly prescribed to minimize menopausal symptoms. MHT is believed to protect the brain and reduce the risk of Alzheimer's disease; however, the evidence supporting this claim is conflicting.

Furthermore, additional research is necessary to evaluate the risks and benefits of MHT on brain health. It remains unclear whether other factors, including a genetic predisposition to Alzheimer's disease, the age at which MHT begins, the formulation, and the route of administration (e.g., pills or a transdermal patch), affect the impact of MHT on the brain. Addressing these questions is crucial to inform clinical decision-making regarding the prescription of MHT and to subsequently enhance women's brain health during and beyond the menopause transition.

Barth et al. analyzed brain imaging data as well as data about lifestyle and demographic factors and histories of gynecological surgeries, of over 20,000 current, past, and never users of MHT from the UK Biobank cohort. As a proxy for brain health, they calculated the so-called 'brain age gap', which refers to the difference between a person's actual chronological age and their estimated brain age calculated from brain imaging data using machine learning; e.g., if someone's chronological age is 50 years and their predicted brain age is 45, the brain age gap is –5 years. A negative brain age gap indicates a 'younger-looking' brain, while a positive brain age gap suggests a brain that appears 'older'. Barth et al. further looked at the volume of the hippocampus, a brain region important for memory, learning and emotion regulation.

They found that women who were current MHT users had an 'older looking brain' than women who had never taken MHT. The volume of the hippocampus was also smaller. In past users, the age at which MHT was last taken also made a difference. Those who were older at the time of their last use after menopause had an "older looking" brain and lower hippocampal volumes. Similar results were found for women who took MHT for longer.

However, women on MHT who had undergone surgery to remove their womb and/or both ovaries had a 'younger looking brain' than women on MHT without similar surgical histories. Interestingly, factors such as a genetic risk for Alzheimer's disease, differences in formulations, or methods of administration did not appear to affect brain health in this study.

The findings of Barth et al. suggest that MHT does not have a general neuroprotective effect, nor does it lead to severe adverse effects on brain health. Instead, the impact of MHT on the brain may be influenced by various factors, including age, duration of use, and past surgical history. However, since the study is cross-sectional, meaning it was conducted at a single point in time, direct causality cannot be established.

Future research examining the long-term impact of MHT on brain health is crucial for understanding individual risk profiles and benefits. Women around the world are faced with critical decisions regarding MHT use, yet the current lack of comprehensive research leaves them without the necessary evidence to make informed choices.

---

According to the 'critical window hypothesis,' MHT might be neuroprotective if it is initiated close to menopause (*Maki, 2013*). For example, MHT initiation during perimenopause has been associated with improved memory and hippocampal function later in life (*Maki et al., 2011*). Although emerging evidence supports this hypothesis (*MacLennan et al., 2006*; *Gibbs and Gabor, 2003*), oral CEE use in combination with MPA has been found to increase the risk for memory decline regardless of timing (*Shumaker et al., 2003*; *Maki, 2013*; *Maki et al., 2007*). Similarly, systemic MHT (i.e. oral and transdermal use) has been associated with a 9–17% increased risk of AD in a Finish nationwide case-control study, independent of MHT formulation and timing (*Savolainen-Peltonen et al., 2019*). However, vaginal estradiol use did not show such risk, indicating differential effects of route of administration (*Savolainen-Peltonen et al., 2019*). Vaginal as well as transdermal

MHT formulations are mainly composed of estradiol, and CEE is a blend of compounds, mostly estrone (*Wharton et al., 2013*). Both estrogens have different affinities to bind to estrogen receptors (ER): relative to estradiol, estrone is approximately 2/3 the affinity to ER-alpha, and about 1/3 to ER-beta (*Kuiper et al., 1997*), and estrone is present at higher levels post-menopause than estradiol. Furthermore, contrary to oral use, vaginal and transdermal MHT formulations bypass hepatic metabolism, resulting in a steady-state concentration of estradiol (estrone:estradiol ratio of 1:1) (*Wharton et al., 2013*). Hence, vaginal, or transdermal estradiol-based MHT formulations might be more effective than oral estrone-based types. For instance, Gleason and colleagues found that females exposed to oral CEE exhibited poorer memory performance than either MHT-naïve or estradiol-exposed individuals (*Gleason et al., 2006*). In addition to estrogens, progestins are commonly added in non-hysterectomized females, and, like estrogens-based MHT, progestins can be administered in different forms, e.g., norethisterone acetate (synthetic progestin), micronized progesterone (bioidentical), or MPA (synthetic progestin). These progestin forms have been linked to different side-effect profiles (*de Lignières, 1999*) and can antagonize the effects of estrogens in MHT (*Nilsen and Brinton, 2002*).

In addition to the impact of MHT timing, formulation, and route of administration, effects of MHT on the female brain might be modulated by apolipoprotein ε type 4 (APOE ε4) genotype. Carried by 14% of the world's population, the APOE ε4 allele is a known dose-dependent risk factor for late-onset AD. Yaffe and colleagues found that among non-carriers, current MHT use lowered the risk of cognitive impairment by almost half compared to never-users, while there was no such effect among carriers (*Yaffe et al., 2000*). Results from the Nurses' Health Study found that MHT use was associated with *worse* rates of decline in general cognition, especially among females with an APOE ε4 allele (*Kim et al., 2019*). Conversely, we found no significant interactions between APOE ε4 genotype and MHT use on cognition in a 2023 UK Biobank study (*Lindseth et al., 2023*). We did, however, observe that earlier MHT initiation was linked to younger GM brain age, albeit weakly, only in APOE ε4 carriers (*Lange et al., 2020*). Moreover, a 2024 UK Biobank study showed smaller brain volumes in MHT users compared to never-users, specifically among females with the APOE ε4/ε4 genotype (*Ambikairajah et al., 2024*). Yet, the potential of APOE ε4 to modulate effects of MHT dosage, administration, and formulation on the female brain is understudied.

In summary, emerging evidence suggests differential effects of MHT formulation, age at initiation, route of administration, and genotype on the female brain and cognition. However, studies aiming to disentangle the effects of different MHT regimes are largely missing (*Kim and Brinton, 2021*). In this observational study, we investigated associations between MHT variables, different MHT regimes, APOE ε4 status, and brain measures in middle to older-aged females from the UK Biobank cohort. MHT variables included user status (i.e. current users, past users, never users), age at initiation, dosage and duration, formulation, route of administration, and type (i.e. bioidentical vs synthetic) as well as active ingredient (e.g. estradiol hemihydrate). MHT regimes based on prescription data were extracted from primary care records (general practice). Brain measures included GM and WM brain age gap based on brain age prediction, hippocampal volumes, and total WM hyperintensity volume as a proxy of vascular disease. These measures were chosen as they have been linked to both chronological and endocrine aging as well as MHT use in our studies (*Lange et al., 2020*; *Ambikairajah et al., 2024*; *Schindler et al., 2023*; *Subramaniapillai et al., 2022*).

## Methods
### Sample characteristics
The sample was drawn from the UK Biobank cohort (https://www.ukbiobank.ac.uk). Females with diffusion- and T1-weighted magnetic resonance imaging (MRI) data and complete data on demographic factors, lifestyle factors, and BMI from the MRI assessment time point were included, yielding a sample of 20,325. Out of this sample, a total of 19,846 participants had complete data related to MHT user status (*Tables 1–2*), and 14,693 participants had complete data on APOE ε4 status. These samples provided the basis for the general MHT-use analyses. Among the MHT users, a subsample of 538 participants had complete MHT-related prescription data and MRI data (*Table 3*, n=521 with data on APOE ε4 status).

**Table 1.** Sample demographics of menopausal hormone therapy (MHT) never-, current-, and past- users in the whole sample.

| | MHT User Status | | | p-value | | |
|---|---|---|---|---|---|---|
| | Never | Current | Past | Never vs Current | Never vs Past | Current vs Past |
| N | 12,012 | 1153 | 6681 | | | |
| Age* | 61.6±7.1 | 60.1±6.8 | 67.5±6.2 | <0.001 | <0.001 | <0.001 |
| Ethnic Background, N (%) | | | | 0.084 | <0.001 | 0.824 |
| White | 11,590 (96.6) | 1128 (98.0) | 6555 (98.2) | | | |
| Asian | 103 (0.9) | 5 (0.4) | 27 (0.4) | | | |
| Black | 96 (0.8) | 4 (0.3) | 23 (0.3) | | | |
| Chinese | 54 (0.5) | 1 (0.1) | 12 (0.2) | | | |
| Mixed | 81 (0.7) | 5 (0.4) | 27 (0.4) | | | |
| Other ethnic group | 73 (0.6) | 8 (0.7) | 28 (0.4) | | | |
| Education, N (%) | | | | 0.260 | <0.001 | <0.001 |
| College/University degree | 6,123 (51.0) | 606 (52.6) | 2,813 (42.1) | | | |
| O levels/GCSEs or equivalent | 2,234 (18.6) | 203 (17.6) | 1,447 (21.7) | | | |
| A levels/AS levels or equivalent | 1,656 (13.8) | 135 (11.7) | 758 (11.3) | | | |
| CSEs or equivalent | 471 (3.9) | 52 (4.5) | 247 (3.7) | | | |
| NVQ/HND/HNC or equivalent | 414 (3.4) | 38 (3.3) | 279 (4.2) | | | |
| Other professional qualifications | 602 (5.0) | 70 (6.1) | 523 (7.8) | | | |
| None of the above | 512 (4.3) | 49 (4.2) | 613 (9.2) | | | |
| Lifestyle score | 1.7±1.2 | 1.8±1.3 | 1.7±1.2 | **0.009** | 0.076 | 0.089 |
| BMI* (m²/kg) | 26.0±4.8 | 25.5±4.4 | 26.2±4.6 | **0.003** | **0.003** | **<0.001** |
| Menopausal Status, N (%) | | | | <0.001 | <0.001 | <0.001 |
| No | 962 (8.0) | 45 (3.9) | 37 (0.6) | | | |
| Yes | 9,905 (82.5) | 769 (66.7) | 5,528 (82.8) | | | |
| Not sure – had a hysterectomy | 545 (4.5) | 247 (21.4) | 1,057 (15.8) | | | |
| Not sure – other reason | 600 (5.0) | 92 (8.0) | 58 (0.9) | | | |
| Oophorectomy, yes, N (%) | 443 (3.7) | 224 (19.4) | 1172 (17.7) | **<0.001** | **<0.001** | 0.159 |
| Hysterectomy, yes, N (%) | 518 (4.5) | 117 (12.5) | 1083 (18.8) | **<0.001** | **<0.001** | **<0.001** |
| APOE ε4 status, carrier, N (%) | 3134 (27.5) | 284 (26.2) | 1557 (24.7) | 0.398 | **<0.001** | 0.278 |
| APOE ε4, allele, N (%) | | | | 0.649 | **<0.001** | 0.460 |
| non-carrier | 8264 (72.5) | 798 (73.8) | 4759 (75.3) | | | |
| ε3/ε4 | 2853 (25.0) | 257 (23.8) | 1424 (22.5) | | | |
| ε4/ ε4 | 281 (2.5) | 27 (2.5) | 133 (2.1) | | | |
| Age started MHT* | | 49.8±6.5 | 47.9±5.6 | | | **<0.001** |
| Age last used MHT* | | 60.1±6.8 | 53.9±6.1 | | | **<0.001** |
| Duration of MHT use* | | 10.3±8.6 | 6.0±5.6 | | | **<0.001** |

*Mean ± Standard Deviation. Age is given in years. Abbreviations: N, sample size; GCSE, General Certificate of Secondary Education; CSE, Certificate of Secondary Education; NVQ, National Vocational Qualification; BMI, body mass index; APOE, apolipoprotein. Significant differences between groups based on t/ χ 2 tests are highlighted in bold.

**Table 2.** Sample demographics of menopausal hormone therapy (MHT) users with and without a history of hysterectomy +/-bilateral oophorectomy in the whole sample.

| | MHT Users | | | p-value | | |
|---|---|---|---|---|---|---|
| | No Surgery | Hysterectomy | Oophorectomy | No vs Hyster | No vs Oopho | Hyster vs Oopho |
| N | 5,510 | 544 | 1,407 | | | |
| Age* | 66.1±6.9 | 69.0±5.6 | 66.7±6.7 | <0.001 | 0.003 | <0.001 |
| Ethnic Background, N (%) | | | | | | |
| White | 5408 (98.3) | 535 (98.5) | 1370 (97.4) | 0.710 | 0.272 | 0.569 |
| Asian | 24 (0.4) | 2 (0.4) | 8 (0.6) | | | |
| Black | 15 (0.3) | 3 (0.6) | 8 (0.6) | | | |
| Chinese | 9 (0.2) | 0 (0.0) | 4 (0.3) | | | |
| Mixed | 22 (0.4) | 2 (0.4) | 6 (0.4) | | | |
| Other ethnic group | 23 (0.4) | 1 (0.2) | 10 (0.7) | | | |
| Education, N (%) | | | | <0.001 | <0.001 | 0.197 |
| College/University degree | 2559 (46.4) | 184 (33.8) | 557 (39.6) | | | |
| O levels/GCSEs or equivalent | 1,099 (19.9) | 132 (24.3) | 323 (23.0) | | | |
| A levels/AS levels or equivalent | 631 (11.5) | 66 (12.1) | 156 (11.1) | | | |
| CSEs or equivalent | 206 (3.7) | 18 (3.3) | 54 (3.8) | | | |
| NVQ/HND/HNC or equivalent | 209 (3.8) | 24 (4.4) | 63 (4.5) | | | |
| Other professional qualifications | 387 (7.0) | 60 (11.0) | 114 (8.1) | | | |
| None of the above | 419 (7.6) | 60 (11.0) | 140 (10.0) | | | |
| Lifestyle score* | 1.7±1.2 | 1.6±1.2 | 1.8±1.2 | 0.323 | 0.066 | 0.044 |
| BMI (kg/m²)* | 25.9±4.50 | 26.1±4.2 | 26.7±4.7 | 0.260 | <0.001 | 0.011 |
| Menopausal Status, N (%) | | | | <0.001 | <0.001 | <0.001 |
| No | 81 (1.5) | 0 (0.0) | 1 (0.1) | | | |
| Yes | 5,290 (96.0) | 444 (81.6) | 615 (43.7) | | | |
| Not sure – had a hysterectomy | 0 (0.0) | 99 (18.2) | 781 (55.5) | | | |
| Not sure – other reason | 139 (2.5) | 1 (0.2) | 10 (0.7) | | | |
| Hysterectomy, yes, N (%) | | | 666 (89.4) | | | |
| APOE ε4 status, carrier, N (%) | 1317 (25.3) | 113 (21.9) | 345 (25.9) | 0.100 | 0.678 | 0.085 |
| APOE ε4, allele, N (%) | | | | 0.095 | 0.427 | 0.161 |
| non-carrier | 3881 (74.7) | 402 (78.1) | 985 (74.1) | | | |
| ε3/ε4 | 1194 (23.0) | 107 (20.8) | 320 (24.1) | | | |
| ε4/ε4 | 123 (2.4) | 6 (1.2) | 25 (1.9) | | | |
| Age at menopause* | 50.0±5.1 | 43.1±6.8 | 46.9±6.3 | <0.001 | <0.001 | <0.001 |
| Age started MHT* | 49.2±5.3 | 46.9±5.7 | 45.6±6.1 | <0.001 | <0.001 | <0.001 |

*Table 2 continued on next page*

*Table 2 continued*

| | MHT Users | | | p-value | | |
|---|---|---|---|---|---|---|
| Age last used MHT* | 54.7±6.1 | 55.0±7.4 | 55.2±7.4 | 0.312 | **0.014** | 0.646 |
| Duration of MHT use* | 5.5±5.4 | 8.1±6.6 | 9.6±7.8 | **<0.001** | **<0.001** | **<0.001** |
| Age MHT rel Age Menopause | | | | **<0.001** | **<0.001** | **<0.001** |
| Same age | 1177 (24.7) | 89 (19.7) | 303 (46.2) | | | |
| After | 1659 (34.8) | 283 (62.7) | 170 (25.9) | | | |
| Before | 1931 (40.5) | 79 (17.5) | 183 (27.9) | | | |
| Age at oophorectomy* | | | 47.7±8.2 | | | |
| Age at hysterectomy* | 44.5±10.0 | 46.9±7.9 | | | | **<0.001** |
| Age Oopho rel Age Menopause | | | | | | |
| Same age | | | 301 (47.8) | | | |
| After | | | 271 (43.0) | | | |
| Before | | | 58 (9.2) | | | |
| Age Oopho rel Age MHT | | | | | | |
| Same age | | | 740 (59.0) | | | |
| After | | | 367 (29.2) | | | |
| Before | | | 148 (11.8) | | | |
| Age Hyster rel Age Menopause | | | | | | **<0.001** |
| Same age | 278 (61.6) | | 324 (54.8) | | | |
| After | 110 (24.4) | | 214 (36.2) | | | |
| Before | 63 (14.0) | | 53 (9.0) | | | |
| Age Hyster rel Age MHT | | | | | | **<0.001** |
| Same age | 80 (17.2) | | 746 (61.2) | | | |
| After | 111 (23.9) | | 296 (24.3) | | | |
| Before | 274 (58.9) | | 177 (14.5) | | | |

*Mean ± Standard Deviation. Age is given in years. Hysterectomy/hyster included females without bilateral oophorectomy; Oophorectomy/Oopho constitutes bilateral oophorectomy (+/-hysterectomy; no hysterectomy n=X, with hysterectomy n=X, with hysterectomy n=Y). Abbreviation: N, sample size; GCSE, General Certificate of Secondary Education; CSE, Certificate of Secondary Education; NVQ, National Vocational Qualification; BMI, body mass index; APOE, apolipoprotein. Significant differences between groups based on $t$ / $\chi 2$ tests are highlighted in bold.

**Table 3.** Sample demographics of menopausal hormone therapy (MHT) users with prescription data, stratified by estrogen-only MHT or combined MHT use.

| | Estrogens-only | Combined | p-value |
|---|---|---|---|
| N | 224 | 314 | |
| Age (years)* | 66.1±6.6 | 65.5±6.7 | 0.318 |
| Education N (%) | | | 0.286 |
| College/University degree | 104 (46.4) | 158 (50.3) | |
| A levels/AS levels or equivalent | 21 (9.4) | 29 (9.2) | |
| O levels/GCSEs or equivalent | 45 (20.1) | 70 (22.3) | |
| CSEs or equivalent | 9 (4.0) | 14 (4.5) | |
| NVQ/HND/HNC or equivalent | 5 (2.2) | 11 (3.5) | |
| Other professional qualifications | 18 (8.0) | 16 (5.1) | |
| None of the above | 22 (9.8) | 16 (5.1) | |
| Lifestyle score* | 1.7±1.2 | 1.8±1.3 | 0.212 |
| BMI (kg/m$^2$)* | 26.1±4.1 | 26.1±4.8 | 0.959 |
| Menopausal Status (%) | | | <0.001 |
| No | 2 (0.9) | 3 (1.0) | |
| Yes | 140 (62.5) | 285 (90.8) | |
| Not sure – had a hysterectomy | 77 (34.4) | 21 (6.7) | |
| Not sure – other reason | 5 (2.2) | 5 (1.6) | |
| APOE ε4 status, carrier, N (%) | 64 (29.4) | 75 (24.8) | 0.284 |
| APOE ε4, allele, N (%) | | | |
| non-carrier | 154 (70.6) | 228 (75.2) | 0.486 |
| ε3/ ε4 | 59 (27.1) | 70 (23.1) | |
| ε4/ ε4 | 5 (2.3) | 5 (1.7) | |
| Hyster-/Oophorectomy, yes (%) | 79 (41.1) | 34 (11.2) | <0.001 |
| Number of drug regimes | 1.39 (0.99) | 2.68 (1.83) | <0.001 |
| Drug dosage, estrogens (mg) | 0.3±0.4 | 1.0±0.7 | <0.001 |
| Drug dosage, progestin (mg) | | 5.4±20.6 | |
| Duration of use, estrogens (weeks) | 202.4±197.7 | 244.8±202.2 | 0.031 |
| Duration of use, progestin (weeks) | | 195.2±174.7 | |
| Route of administration, N (%) | | | <0.001 |
| oral | 40 (17.9) | 193 (61.5) | |
| transdermal | 50 (22.3) | 14 (4.5) | |
| vaginal | 109 (48.7) | 0 (0.0) | |
| injection¹ | 4 (1.8) | 5 (1.6) | |
| mixed | 21 (9.4) | 102 (32.5) | |
| Estrogens, active ingredient, N (%) | | | |
| estradiol hemihydrate° | 141 (62.9) | | |
| CEE | 18 (8.0) | | |
| estradiol° | 34 (15.2) | | |

*Table 3 continued on next page*

*Table 3 continued*

| | Estrogens-only | Combined | p-value |
|---|---|---|---|
| estradiol valerate° | 4 (1.8) | | |
| tibolone | 0 (0.0) | | |
| mixed | 27 (12.1) | | |
| Estrogens + Progestins, active ingredient, N (%) | | | |
| estradiol hemihydrate & norethisterone acetate [1] | | 51 (16.2) | |
| estradiol hemihydrate & dydrogesterone° [2] | | 14 (4.5) | |
| estradiol hemihydrate & norethisterone [1] | | 13 (4.1) | |
| estradiol hemihydrate & levonorgestre [2] | | 2 (0.6) | |
| estradiol hemihydrate & drospirenone [3] | | 1 (0.3) | |
| CEE & norgestrel [2] | | 19 (6.1) | |
| CEE & medroxyprogesterone acetate [1] | | 10 (3.2) | |
| CEE & norethisterone [1] | | 2 (0.6) | |
| estradiol valerate & norethisterone [1] | | 8 (2.5) | |
| estradiol valerate & levonorgestrel [2] | | 3 (1.0) | |
| estradiol valerate & medroxyprogesterone acetate [1] | | 3 (1.0) | |
| estradiol & norethisterone acetate [1] | | 7 (2.2) | |
| estradiol & norethisterone [1] | | 3 (1.0) | |
| estradiol & progesterone° [1] | | 2 (0.6) | |
| tibolone | | 13 (4.1) | |
| mixed | | 163 (51.9) | |

[*]Mean ±Standard Deviation. °Bioidentical form (no circle indicates synthetic form); [1-3]=progestin generations; [1]incl. subcutaneous and intravenous infections. Abbreviations: N, sample size; GCSE, General Certificate of Secondary Education; CSE, Certificate of Secondary Education; NVQ, National Vocational Qualification; BMI, body mass index; APOE, apolipoprotein; CEE, conjugated equine estrogen. Significant differences between groups based on t/ $\chi$ 2 tests are highlighted in bold.

## MRI data acquisition and processing

A detailed overview of the UK Biobank neuroimaging data acquisition and protocols has been published elsewhere (*Alfaro-Almagro et al., 2018*; *Miller et al., 2016*). Harmonized analysis pipelines were used to process raw T1-weighted MRI data for all participants (N=20,540), including automated surface-based morphometry and subcortical segmentation (FreeSurfer, v5.3). To remove poor-quality data likely due to motion, participants with Euler numbers ≥4 standard deviations (SD) (*Rosen et al., 2018*) below the mean were excluded (n=180), yielding a total of 20,360 participants with T1-weighted MRI data (see *Sample characteristics* for final sample size).

In addition to the classic set of subcortical and cortical summary statistics from FreeSurfer (*Fischl et al., 2002*), we utilized a fine-grained cortical parcellation scheme (*Glasser et al., 2016*) to extract cortical thickness, area, and volume for 180 regions of interest per hemisphere. This yielded a total set of 1,118 structural brain imaging features (360/360/360/38 for cortical thickness/area/volume, as well as cerebellar/subcortical and cortical summary statistics, respectively), that were used to predict global GM brain age. The T1-weighted MRI data was residualized with respect to scanning site and intracranial volume using linear models (*Voevodskaya et al., 2014*). To probe hippocampal-specific effects separately (*Duarte-Guterman et al., 2015*), we used the extracted measures of left and right hippocampus volume as additional outcome measures in subsequent analyses.

Diffusion-weighted MRI data were processed using an optimized diffusion pipeline (*Maximov et al., 2019*; *Maximov et al., 2021*; *Voldsbekk et al., 2021*). Metrics from four diffusion models were utilized to predict global WM brain age (see details **Appendix Note 1**). In total, 98 diffusion features

were included (global mean values + tract values for each diffusion metric). The diffusion-weighted data passed TBSS post-processing quality control using the YTTRIUM algorithm (*Maximov et al., 2021*) and were residualized with respect to scanning sites using linear models.

Total volume of WMH was derived from T1-weighted and T2-weighted images using BIANCA (FSL, v6.0); a fully automated, supervised method for WMH detection (*Griffanti et al., 2016*). Preprocessed volumes per participant were exported from the UK Biobank, Field ID: 25781, and log-transformed due to a left-skewed distribution. A total of 19,538 females had data to compute WMH.

## Brain-age prediction

In line with our previous studies (*Voldsbekk et al., 2021*; *Schindler et al., 2022*), tissue-specific age prediction models in females only were run using *XGBoost* regression, which is based on a decision-tree ensemble algorithm (https://github.com/dmlc/xgboost; *XGBoost, 2025*). Hyper-parameters were tuned in nested cross-validations using five inner folds for randomized search and 10 outer folds for model validation. Predicted age estimates were derived using the Scikit-learn library (https://scikit-learn.org), and brain age gap (BAG) values were calculated for each model (predicted – chronological age) to provide estimates of global GM BAG based on T1-weighted data, and global WM BAG based on diffusion-weighted data. The age prediction models were run without a subsample with ICD10 diagnosis known to impact the brain (n=1739, for details see **Appendix Note 2**), and then applied to the respective group of participants with diagnoses to obtain brain age estimates for the whole sample. This approach was selected to base the prediction models on normative age trajectories, while also including a more representative total sample (females both with and without diagnoses) in the subsequent analyses.

## MHT-related variables

For the whole sample, general MHT data included user status (never-user, current-user, or past-user), age first started using MHT, age last used MHT, and duration of MHT use (age last used – age first used). In current MHT users, age at last use was set to their age at the imaging assessment to calculate duration of use.

For a sub-sample, prescription MHT data was extracted from primary care records (general practice, UK Biobank Field ID: 42039) using freely available code (*Anatürk et al., 2023*). We extracted formulation (i.e. estrogens-only, estrogens + progestins), route of administration (i.e. oral, transdermal, vaginal, injection), and daily drug dosage (mg) from the trade names/active ingredients indicated by the treating general practitioner and duration of use (weeks) was calculated based on prescription dates. Trade names were verified using the UK Product Compendium (https://www.medicines.org.uk/emc). MHT formulations were further separated into bioidentical & synthetic, and progestins were stratified by generation (i.e. first to third generation), based on their chemical structures, receptor binding properties, and clinical characteristics (*Sitruk-Ware, 2006*). Although all hormones in MHTs are chemically synthesized, bioidentical hormones are structurally identical to the hormones naturally produced in the human body, whereas synthetic hormones are not. It is debated whether bioidentical and synthetic hormones might have different risk profiles (*Holtorf, 2009*).

Drug duplicates, meaning the same drug issued on the same date per participant, were excluded. Individuals who switched between regimes were labeled as mixed (see *Table 3*). Individuals who took MHT after the imaging assessment, based on MHT prescription dates, were excluded (n=6). In total, 538 participants had both detailed MHT and imaging data, and complete data on key demographic variables such as age, education, and menopausal status (see *Table 3*). Females who had missing data, or had responded 'do not know,' 'prefer not to answer,' 'none of the above' or similar for any of the relevant variables were excluded for the respective analyses. If possible and appropriate, missing data at the imaging time point was replaced with valid data from the baseline assessment. This was the case for the following variables: MHT user status, history of hysterectomy and/or bilateral oophorectomy, and menopausal status. For details on MHT-related variables in the UK Biobank see *Supplementary file 1*.

## APOE ε genotyping

For genotyping, we used the extensive quality control UK Biobank version 3 imputed data (*Bycroft et al., 2018*). The APOE ε genotype was approximated based on the two APOE ε single-nucleotide

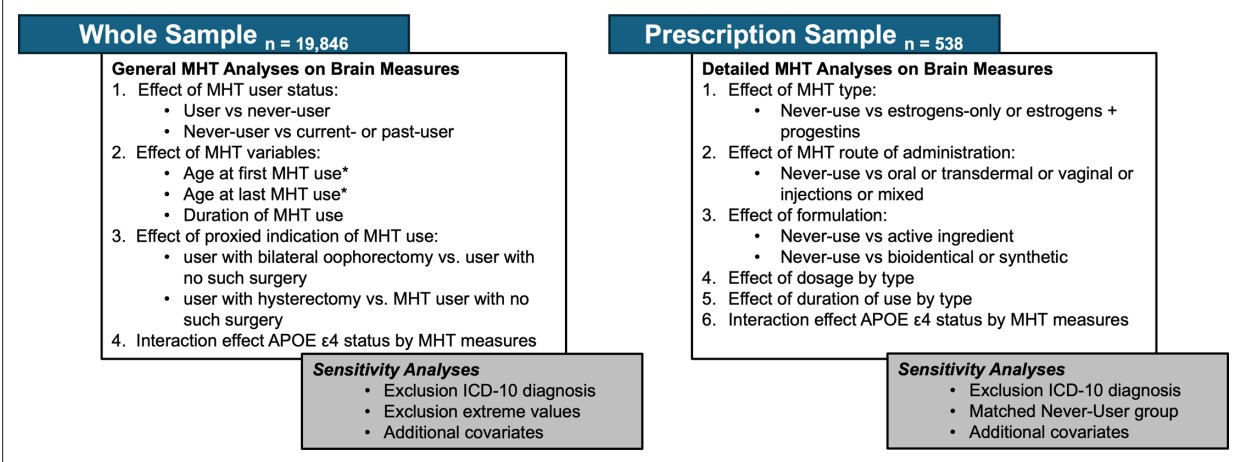

**Figure 1.** Overview of statistical main and sensitivity analyses by sample and research question.

polymorphisms – rs7412 and rs429358 (**Lyall et al., 2016**). APOE e4 status was labeled *carrier* for ε3/ε4 and ε4/ε4 combinations, and *non-carrier* for ε2/ε2, ε2/ε3, and ε3/ε3 combinations. Due to its ambiguity with ε1/ε3, the homozygous ε2/ε4 allele combination was removed (n=484) (**Seripa et al., 2007**) (https://www.snpedia.com/index.php/APOE). Further information on the genotyping process is available in the UK Biobank documentation.

## Statistical analyses

The statistical analyses were run using R, v4.2.2. p-values were corrected for multiple comparisons using false discovery rate (FDR) (**Benjamini and Hochberg, 1995**) correction across all brain measures for all sets of analyses per model (1-3). The sets of FDR corrections are reflected in the corresponding results tables. All variables were standardized prior to performing the regression analyses (subtracting the mean and dividing by the standard deviation). All statistical main and sensitivity analyses by sample and research questions are summarized in **Figure 1**.

## Associations between MHT variables and brain measures in the whole sample

First, we tested for associations between MHT user status (never-user/user) and GM and WM BAG, left and right hippocampus volume, and WMH volume. We further tested whether current and past MHT use, relative to never-use, was associated with the brain measures.

Second, among all MHT users, we assessed associations between age at first use as well as duration of use and brain measures. Among past MHT users, we tested for associations between age at last use and brain measures. In postmenopausal MHT users, we also tested whether age at first and last use in relation to age at menopause (i.e. age started MHT – age at menopause; age last use MHT – age at menopause, respectively) was associated with the brain measures. The following regression models (*lm* function) were fitted for these analyses, with DV representing each MRI measure (i.e. GM BAG, WM BAG, left and right hippocampus volumes, WMH volume) as dependent variable and IV representing each MHT variable (i.e. MHT user status, age at first use, age at last use, duration of use), as independent variable:

$$DV \sim IV + age + education + lifestyle\ score + BMI + menopause\ status \qquad (1)$$

The chosen covariates are known to influence MHT use and brain structure (**Beck et al., 2022**; **de Lange and Cole, 2020**; **Cox et al., 2018**; **Ho et al., 2011**). The lifestyle score was calculated using a published formula (**Foster et al., 2018**), and included data on sleep, physical activity, nutrition, smoking, and alcohol consumption (see **Appendix Note 3, Supplementary file 2**). For these analyses, participants with a history of hysterectomy and/or bilateral oophorectomy were excluded (n=3903), as they might have an increased risk for neurological decline (**Phung et al., 2010**). Note that Model 1 was not adjusted for menopause status in the analyses only including postmenopausal MHT users.

To test for differences in brain measures between MHT users with a history of hysterectomy without bilateral oophorectomy or bilateral oophorectomy (+/-hysterectomy, proxy of surgical menopause) relative to MHT users without such surgeries, we ran additional regression models within the sample of MHT users using the same model as specified above (model 1).

## Associations between MHT variables and brain measures in a subsample with prescription MHT data

First, we tested whether MHT formulation (i.e. estrogens-only, estrogens + progestins, none) and route of administration (i.e. oral, transdermal, vaginal, injections, mixed, none) was associated with brain measures. Never-users (dummy-coded as 'none') served as a reference group. The following regression model was fitted for these analyses, with DV representing each MRI measure and IV representing MHT formulation or route of administration:

$$DV \sim IV + age + education + menopause\ status \tag{2}$$

For these analyses, we only included age, education, and menopause status as covariates to retain the largest possible sample size (n=538; see 'Sensitivity Analyses' for follow-up analyses including history of hysterectomy and/or bilateral oophorectomy as an additional covariate).

In estrogens-only users, we tested whether different active ingredients (i.e. estradiol valerate, estradiol hemihydrate, CEE, estradiol, mixed, none) and bioidentical or synthetic estrogens forms were associated with brain measures, relative to never-users as dummy-coded reference group. We further assessed whether duration of estrogens use (weeks), and estrogens drug dosage was associated with brain measures among estrogens-only users. We used the same model as specified above (model 2).

In estrogens + progestin users, we also explored whether different active ingredients were associated with brain measures, relative to never-users (see model 2). For this analysis, we only included formulations with N ≥ 10 (see *Table 3*). We further tested whether bioidentical or synthetic MHT or a mix of both, and progestin generations (i.e. first or second) were associated with brain measures, relative to never-users (see model 2). Third progestin generation users (n=1) were excluded. Among estrogens + progestin users, we further assessed whether duration of estrogens/progestin use (weeks) and drug dosage of estrogens/progestin was differently associated with brain measures. For these models, the estrogens/progestin measures (i.e. duration of use or drug dosage) were included in the same model. The covariates specified in model 2 were included.

## Associations between MHT variables and brain measures by APOE ε4 status

First, we tested for the main effects of APOE ε4 status (i.e. carrier vs non-carrier) and APOE ε4 dose (i.e. non-carrier, ε3/ε4, ε4/ε4) on brain measures, in separate models:

$$DV \sim APOE4 + age + education + lifestyle\ score + BMI + menopause\ status \tag{3}$$

Similar to model 1, participants with a history of hysterectomy and/or bilateral oophorectomy were excluded for these analyses. Non-carrier served as a reference group.

Second, we re-ran models 1–2 including an APOE ε4 status × MHT measure interaction term to assess the effects of APOE ε4 status on the associations between MHT variables and brain measures. Main effects for the interaction terms were automatically included in the model.

### Sensitivity analyses

To test whether the results were influenced by the inclusion of participants with ICD-10 diagnosis or by non-linear effects of age, the main analyses (models 1–2) were re-run excluding the sub-sample with diagnosed brain disorders (see **Appendix Note 2**) or adding age (*Ding et al., 2013*) as additional covariate, respectively. In addition, we re-ran the analyses for MHT formulation and route of administration (model 2) adjusting for a history of hysterectomy and/or bilateral oophorectomy in addition to age, BMI, and lifestyle score (n=460). Since estrogens-only MHTs are commonly prescribed after hysterectomy ±bilateral oophorectomy, we included surgical history as a covariate instead of excluding participants with a surgical history.

To adjust for the potential influence of extreme values on our results, we assessed each continuous MHT variable (i.e. age at first use, age at last use, age at menopause) for extreme values using a

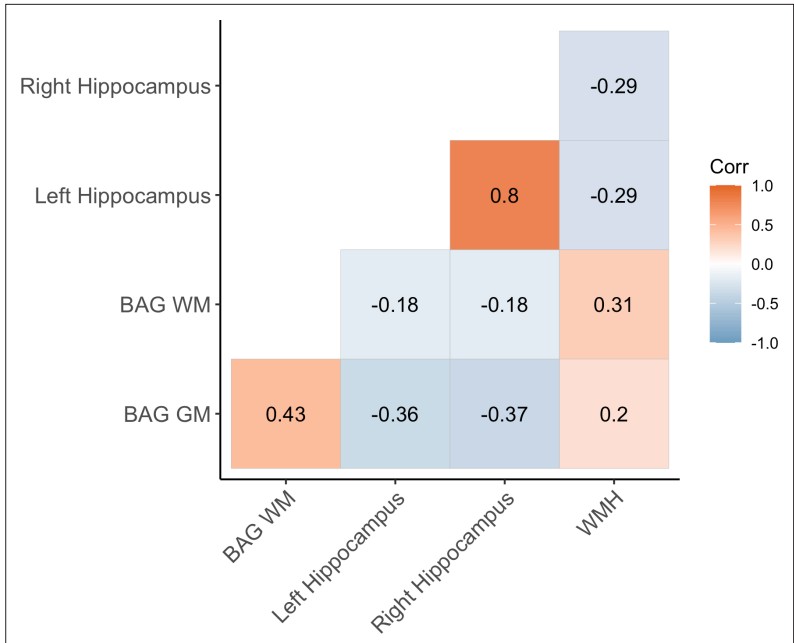

**Figure 2.** Correlations (Pearson's r) between white matter (WM) and gray matter (GM) brain age gap (BAG) as well as left and right hippocampal volumes and total white matter hyperintensity (WMH) volume. BAG measures are adjusted for age (*Seripa et al., 2007*). WM BAG, GM BAG, and hippocampal volumes were available for 20,360 individuals, and 19,538 had data on WMH volume.

data-driven approach and excluded the corresponding participants before re-running the respective main analyses (model 1). To identify extreme values, we applied the median absolute deviation (MAD) method (R package *Routliers*), using default settings (i.e. a MAD threshold of ±3).

For relevant analyses, the subsample with prescription MHT data was compared to all available never-users, to allow for a large and representative control group. However, to assess whether the results were sensitive to the control group selection, we matched the prescription MHT data sample (n=538) to an equally sized subsample of never-users (n=538), using genetic matching without replacement (*matchit* R package). Genetic matching is a form of nearest neighbor matching where distances are computed as the generalized Mahalanobis distance. The generalization of the Mahalanobis distance is achieved with a scaling factor for each covariate that represents the importance of that covariate to the distance. The groups were matched based on the covariates used in model 2, namely age, education, and menopause status. In the matched sample, for analyses comparing users relative to never-users, model 2 was rerun adjusting only for age.

## Results

### Sample characteristics

Sample demographics including lifestyle score, stratified by MHT user group, surgical history among MHT users, and estrogen-only MHT or combined MHT use, are summarized in *Tables 1–3*, respectively. MHT user group differences for each lifestyle factor contained in the lifestyle score are shown in *Supplementary file 2*.

### Brain age prediction

The age prediction accuracy largely corresponded to our previous UK Biobank studies in overlapping samples (*Lange et al., 2020*; *Schindler et al., 2022*), as shown in *Supplementary file 3*. *Figure 2* shows the correlations between GM BAG, WM BAG, left and right hippocampus volume, and WMH volume.

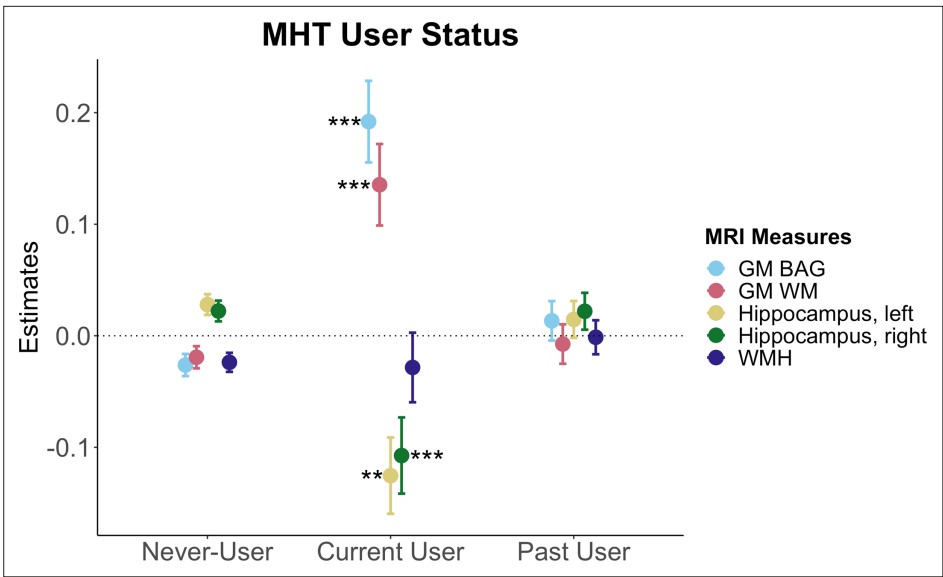

**Figure 3.** Associations between brain magnetic resonance imaging (MRI) measures and menopausal hormone therapy (MHT) user status. Point plot of estimated marginal means with standard errors from separate regression models with brain measure as dependent variable and MHT user status as independent (categorical) variable, with never-users as a reference group. Brain measures include white matter (WM) and gray matter (GM) brain age gap (BAG) as well as left and right hippocampal volumes and total white matter hyperintensity (WMH) volume. Sample sizes per user group, excluding participants with a history of hysterectomy and/or bilateral oophorectomy, were as follows: never-users (n = 10,934), current user (n = 802), past user (n = 3,658). The models were adjusted for age, education, body mass index, lifestyle score, and menopausal status. All variables were standardized prior to performing the regression analysis (subtracting the mean and dividing by the standard deviation). Stars indicate significant associations. Significance codes: 0 '***' 0.001 '**' 0.01 '*'.

## Associations between MHT variables and brain measures in the whole sample

In the whole sample, MHT users showed higher GM BAG (i.e. older brain age relative to chronological age; $\beta$ = 0.034, p = 6.74e-05, $p_{FDR}$ = 0.001) and lower left hippocampal volume ($\beta$ = −0.02, p = 0.006, $p_{FDR}$ = 0.02) compared to never-users (*Supplementary file 4*). As shown in *Figure 3*, stratifying the sample into never-, past-, and current-users showed statistically significant higher GM and WM BAG and lower left and right hippocampal volumes in current users compared to never-users, but no significant difference in past-users relative to never-users. For WMH volume, there were no significant differences between current and past MHT users relative to never-users, respectively (*Supplementary file 4*, *Figure 3*).

Among MHT users, we found no relationships between age at MHT initiation, alone and in relation to age at menopause, and the MRI variables (*Supplementary file 4*). However, in past MHT users, older age at last use was associated with higher GM BAG. In postmenopausal past MHT users, older age at last use after age at menopause was associated with higher GM and WM BAG, higher WMH volume, and lower left and right hippocampal volumes. Similarly, longer duration of MHT use was associated with higher GM BAG and WM BAG, as well as lower left and right hippocampal volumes (*Supplementary file 4*).

MHT use with a history of hysterectomy ± bilateral oophorectomy was associated with *lower* GM BAG (i.e. younger brain age relative to chronological age) relative to MHT users without such history. In addition, hysterectomy without oophorectomy was associated with higher left and right hippocampus volumes relative to MHT users without surgical history (*Figure 4*).

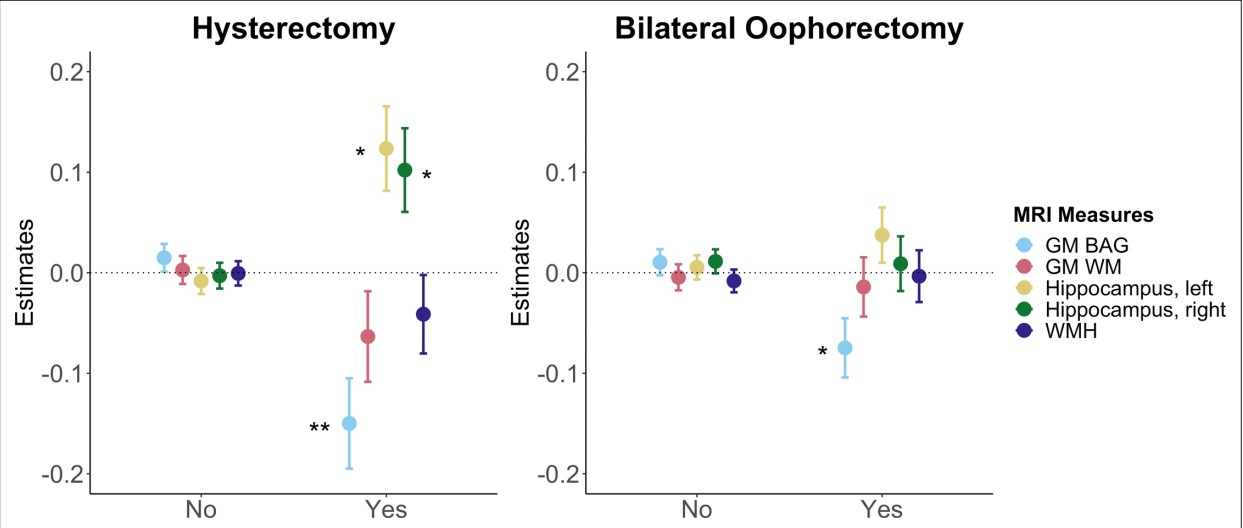

**Figure 4.** Associations between brain magnetic resonance imaging (MRI) measures and history of hysterectomy and/or bilateral oophorectomy in menopausal hormone therapy (MHT) users. Point plot of estimated marginal means with standard errors from separate regression analysis with MRI measure as dependent variable and history of hysterectomy and/or bilateral oophorectomy as independent (categorical) variable. MHT users without such surgical history served as a reference group. Brain measures include white matter (WM) and gray matter (GM) brain age gap (BAG) as well as left and right hippocampal volumes and total white matter hyperintensity (WMH) volume. Sample sizes were as follows: hysterectomy only (no, n = 5522; yes, n = 546) and bilateral oophorectomy +/- hysterectomy (no, n = 6513; yes, n = 1412). The models were adjusted for age, education, body mass index, lifestyle score, and menopausal status. All variables were standardized prior to performing the regression analysis (subtracting the mean and dividing by the standard deviation). Stars indicate significant associations. Significance codes: 0 '***' 0.001 '**' 0.01 '*'.

## Associations between MHT variables and brain measures in a subsample with prescription MHT data

We found no significant associations between MHT formulation, route of administration, MHT type (i.e. bioidentical vs synthetic), MHT active ingredients, progestin generation, dosage, duration of use, and brain measures after adjusting for multiple comparisons (*Supplementary file 5*).

Before adjusting for multiple comparisons, we observed higher WM BAG in estrogens + progestin users relative to never-users ($\beta$ = 0.127, p = 0.026, $p_{FDR}$ = 0.832). Users of CEE + MPA ($\beta$ = 0.634, p = 0.045, $p_{FDR}$ = 0.994) as well as mixed active ingredients ($\beta$ = 0.179, p = 0.023, $p_{FDR}$ = 0.832) showed higher WM BAG compared to never-users, and oral users showed higher WMH volume ($\beta$ = 0.134, p = 0.020, $p_{FDR}$ = 0.832). We further observed an association between longer duration of estrogens use and lower WMH volume in estrogens-only users relative to never-users ($\beta$ = −0.146, p = 0.028, $p_{FDR}$ = 0.832). In estrogens + progestin users, longer duration of estrogens use was trend-level associated with lower left and right hippocampal volumes (left: $\beta$ = −0.149, p = 0.031, $p_{FDR}$ = 0.832; right: $\beta$ = −0.165, p = 0.015, $p_{FDR}$ = 0.832).

## Associations between MHT variables and brain measures by APOE ε4 status

We found significantly lower right hippocampal volumes in carriers of two APOE ε4 alleles compared to non-carriers (*Figure 5*). We also observed higher GM BAG and lower left hippocampus volumes in APOE ε4-carriers relative to non-carriers (*Supplementary file 6*), but these effects were not significant after adjusting for multiple comparisons. We found no significant interactions between APOE ε4 status and MHT variables on brain measures after adjusting for multiple comparisons, both for the analyses including the whole sample (*Supplementary file 7*) and for the MHT prescription sample (*Supplementary file 8*).

Before adjusting for multiple comparisons, we observed a trend-level interaction between APOE ε4 status and MHT use in participants with a hysterectomy without oophorectomy on WMH volume ($\beta$ = −0.025, p = 0.046, $p_{FDR}$ = 0.927) in the whole sample. In the prescription sample, we observed several interactions between APOE ε4 status and MHT variables on brain measures before FDR correction. For example, mixed estrogens-only use showed an interaction with APOE ε4 status on right hippocampus

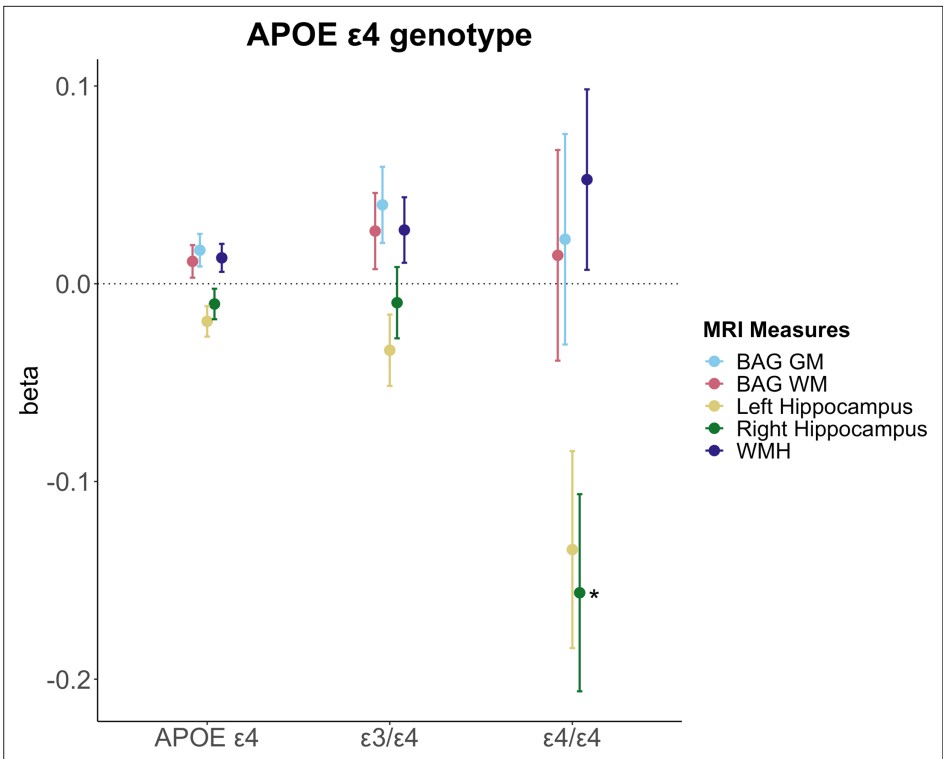

**Figure 5.** Associations between apolipoprotein ε type 4 (APOE ε4) genotype and brain magnetic resonance imaging (MRI) measures. Point plot of standardized beta-values with standard errors from separate multiple regression analysis with MRI measure as dependent variable and APOE ε4 genotype as independent variable. Non-carrier served as a reference group (n=10,787). APOE ε4 (all, n=3935) represents the ε3/ε4 (n=3572) and ε4/ε4 carriers (n=363) grouped together. For WMH, the sample sizes were as follows: non-carrier (n=10,377), APOE ε4 (all, n=3410), ε3/ε4 (n=3410), and ε4/ε4 carriers (n=349). Brain measures include white matter (WM) and gray matter (GM) brain age gap (BAG) as well as left and right hippocampal volumes and total white matter hyperintensity (WMH) volume. The models were adjusted for age, education, body mass index, lifestyle score, and menopausal status. All variables were standardized prior to performing the regression analysis (subtracting the mean and dividing by the standard deviation). Stars indicate significant associations. Significance codes: 0 '***' 0.001 '**' 0.01 '*'.

volume ($\beta$ = −0.491, p = 0.032, $p_{FDR}$ = 0.511), indicative of smaller volumes in APOE ε4 carriers relative to non-carriers. We also observed an interaction between APOE ε4 status and synthetic estrogens + progestin use on GM BAG ($\beta$ = −0.512, p = 0.020, $p_{FDR}$ = 0.449), left hippocampus volume ($\beta$ = 0.502, p = 0.016, $p_{FDR}$ = 0.449), and WMH volume ($\beta$ = −0.412, p = 0.029, $p_{FDR}$ = 0.449), suggesting lower GM BAG, larger left hippocampus volume, and lower WMH volume in APOE ε4 carriers compared with non-carriers. Please see **Appendix Note 4** for the results of all models including APOE ε4 in the prescription sample.

## Sensitivity analyses

The results were consistent after removing participants with ICD-10 diagnoses known to impact the brain (see *Supplementary file 9* for model 1 analyses and *Supplementary file 10* for model 2 analyses), after additionally adjusting for age (see *Supplementary file 11*), and after removing extreme values (see *Supplementary file 12* for model 1 analyses). Detected extreme values are highlighted in *Supplementary file 13*. Similarly, additionally adjusting for BMI, lifestyle score, and history of hysterectomy and/or bilateral oophorectomy (model 2, *Supplementary file 14*) or matching never-users to MHT users did not alter the results for the prescription dataset (model 2, *Supplementary file 15*).

## Discussion

This study assessed detailed MHT data, APOE ε4 genotype, and brain characteristics in a large, population-based sample of females in the UK. The results showed significantly higher GM and WM BAG (older brain age relative to chronological age) as well as smaller left and right hippocampus volumes in current MHT users, but not in past users, compared to never-users. Among MHT users, we found no significant associations between age at MHT initiation, alone and in relation to age at menopause, and brain measures. However, older age at last use after age at menopause was associated with higher GM and WM BAG, higher WMH volume, and lower left and right hippocampal volumes. Similar associations were found for longer duration of MHT use. MHT users with a history of hysterectomy ± bilateral oophorectomy showed *lower* GM BAG relative to MHT users without such history. In the sub-sample with prescription data, we found no significant associations between detailed MHT variables, such as MHT formulation, route of administration, type, active ingredients, and dosage, and brain measures, after FDR correction. Lastly, although we found lower right hippocampus volumes in carriers of two APOE ε4 alleles relative to non-carriers, we found no interactions with MHT variables after FDR correction.

Current MHT users showed higher GM and WM BAG as well as lower left and right hippocampus volumes relative to never-users. The effects were robust but relatively modest in magnitude, with the largest effect size indicating a group difference of 0.77 y (~9 mo) for GM BAG (standardized $\beta$ = 0.218, *Supplementary file 4*). However, we found no significant differences in brain measures in past MHT users relative to never-users. Current MHT users were significantly younger than past- and never-users, and around 67% were menopausal relative to over 80% in the past- and never-user groups. The unequal distribution of age and menopausal status across groups may have influenced the observed findings. For instance, a larger proportion of the current users might be in the perimenopausal phase, which is often associated with debilitating neurological and vasomotor symptoms (*Brinton et al., 2015*). MHT is commonly prescribed to minimize such symptoms. Although MHT initiation during perimenopause has been associated with improved memory and hippocampal function, as well as lower AD risk later in life (*Maki et al., 2011*), the need for MHT might in itself be an indicator of neurological changes (*Kantarci and Manson, 2023*) here potentially reflected in higher BAG and lower hippocampal volumes. After the transition to menopause, symptoms might subside and some perimenopausal brain changes might revert or stabilize in the postmenopausal phase (*Mosconi et al., 2021*). Although the UK Biobank lacks detailed information on menopausal symptoms and perimenopausal staging, our results might be capturing subtle disturbances during perimenopause that later stabilize. This could explain why the largely postmenopausal groups of past MHT users and never-users present with lower GM and WM BAG than the current user group. Considering the critical window hypothesis emphasizing perimenopause as a key phase for MHT action (*Maki, 2013*; *Kim and Brinton, 2021*), future longitudinal studies are crucial to clarify the interplay between neurological changes and MHT use across the menopause transition.

Besides age-related differences, the current user group also showed a significantly unhealthier lifestyle, and higher rates of bilateral oophorectomy compared to the never-user group and the past-user group, respectively. These findings are contrary to the healthy user bias hypothesis (i.e. equating MHT use with healthy user status, *Gleason et al., 2012*), and might indicate that the current user group could be exposed to neurological changes prior to MHT use. According to the healthy cell bias hypothesis of estrogen action (*Brinton, 2008*), MHT use might be detrimental for brain health when initiated after cells are exposed to neurological degeneration. Although MHT use might have exacerbated adverse brain changes in the unhealthier group of current users, higher BAG was also linked to longer duration of MHT use and older age at last use post-menopause. Although the effect sizes were modest (*Supplementary file 4*), these findings might reflect subtle yet unfavorable effects of MHT on brain health in our sample, particularly when used continuously after natural menopause with uterus and ovaries still intact (i.e. all females with hysterectomy ± bilateral oophorectomy were excluded from this analysis).

On the contrary, MHT users with a history of hysterectomy ± bilateral oophorectomy showed *lower* GM BAG relative to MHT users without such history, and we found larger hippocampal volumes in MHT users with a hysterectomy without a bilateral oophorectomy. Previous work associates these surgical interventions with accelerated cognitive decline (*Rocca et al., 2007*), increased risk of dementia (*Phung et al., 2010*), morphological changes in regions of the medial temporal lobe (*Zeydan et al.,*

2019), and accumulation of Alzheimer's disease pathology (*Georgakis et al., 2019*). However, some of these studies relied on comparing surgery to no surgery (*Phung et al., 2010*; *Rocca et al., 2007*) without taking MHT use into account (*Georgakis et al., 2016*), or investigated the effect of surgery with MHT relative to surgery without MHT. For instance, a 2023 study highlighted that estrogen-only use in females with hysterectomy was associated with increased dementia rates relative to females with hysterectomy who never used MHT (*Pourhadi et al., 2024*). In the current study, we specifically compared the effect of MHT use on brain measures among females with and without surgical history and found lower GM BAG in the surgical MHT user group. These findings might be explained by differences in indication of MHT use for females with and without surgery, MHT formulation, and age at surgery. Without surgery, MHT use (i.e. often combined MHT formulation) is prescribed in females with intact uterus and ovaries to minimize menopausal symptoms that are largely neurological in nature, such as vasomotor symptoms as well as mood, cognitive, and sleep disturbances (*Brinton et al., 2015*). After hysterectomy, estrogen-only MHT might be prescribed but is not strictly needed as ovaries are still intact. Whereas after bilateral oophorectomy (=surgical menopause), combined or estrogen- only MHT is indicated to compensate for the acute and chronic deficiency of hormones normally produced by the ovaries. These distinctions might entail important differences in risk and side-effect profiles of estrogen-only or combined MHT in females with and without endogenous sex hormone production. It is also possible that the timing between MHT use and surgery is more tightly controlled and, therefore, more beneficial for brain aging (*Kim and Brinton, 2021*). For instance, studies suggest that MHT may mitigate the potential long-term adverse effects of bilateral oophorectomy before natural menopause on bone mineral density as well as cardiovascular, cognitive, and mental health (*Blümel et al., 2022*; *Kaunitz et al., 2021*; *Stuursma et al., 2022*). In addition, a 2024 UK Biobank study found that ever used MHT was associated with decreased odds of Alzheimer's disease in women with bilateral oophorectomy (*Calvo et al., 2024*).

However, our study also showed group differences in several demographic and MHT-related variables which might influence the results. For instance, the hysterectomy MHT user group was significantly older than the non-surgery and bilateral oophorectomy group, started MHT at a younger age than the non-surgery group, and had surgery at a younger age than the bilateral oophorectomy group. More research is needed to disentangle this complex interplay of MHT effects in females with and without intact uteri and ovaries.

No significant associations were observed between brain measures and MHT regimes based on prescription data. However, given the relatively small sample size and the large number of comparisons, it is possible that we were unable to detect subtle effects of factors such as MHT formulation, route of administration, type, active ingredients, and dosage. Before FDR correction, we found higher WM BAG in estrogens +progestin users, in CEE + MPA users, and in mixed active ingredients users relative to never-users. In addition, WMH volume was higher among oral MHT users and lower with longer duration of estrogens use in estrogens-only users relative to never-users. These uncorrected results are partly in line with previous findings. For instance, the 2003 Women's Health Initiative Memory Study reported that prolonged oral use of both CEE alone (*Shumaker et al., 2004*), or combined with MPA (*Shumaker et al., 2003*), increased the risk of dementia and cognitive decline among females aged 65 y and older. Contrarily, prolonged use of estrogen-only MHT has been linked to reduced white matter loss in aging (n=10) (*Ha et al., 2007*). Similar to our findings, Ha and colleagues did not find an association between gray matter volumes and estrogen use (*Ha et al., 2007*). Although uncorrected results must be interpreted with caution, our findings might indicate an unfavorable effect of mixed active ingredient use, including CEE ± MPA and oral administration, on white matter brain aging. This effect might be due to the higher estrone concentrations associated with such oral estrone-based MHT types (*Longcope et al., 1985*; *Slater et al., 2001*; *Van Erpecum et al., 1991*). In addition, over 50% of combined MHT users had mixed MHT use which suggests that it might be beneficial to examine the reasons for switching medication. In sum, these findings highlight the need for personalized MHT regimes, and longitudinal RCTs are needed to establish how different MHT regimes, and their respective hormonal profiles are causally linked to brain aging.

In the current study, we found significantly smaller right hippocampus volumes in carriers of two APOE ε4 alleles relative to non-carriers, which is in line with previous work linking the APOE ε4 genotype to greater rates of hippocampal atrophy in non-demented and Alzheimer's disease samples (*Saeed et al., 2021*; *Gorbach et al., 2020*). However, after FDR correction, we did not find any

interactions between APOE ε4 carrier status and MHT variables in relation to the brain measures. This finding was unexpected, as previous work has highlighted the APOE ε4 genotype as a crucial determinant of MHT effects on the female brain (*Barth et al., 2023*). For instance, one study associated MHT use with improved memory performance and larger entorhinal and amygdala volumes in female ε4 carriers versus non-carriers (*Saleh et al., 2023*). A 2023 prospective longitudinal study showed that MHT was associated with smaller changes towards Alzheimer's disease pathophysiology than non-therapy and that APOE ε4 carrier status was linked to an amplified treatment outcome (*Depypere et al., 2023*). However, contrary results have also been reported. For instance, Yaffe and colleagues found lower risk of cognitive decline with estrogen-only MHT use among female APOE ε4 non-carriers, but there was no such effect among carriers (*Yaffe et al., 2000*). To understand these discrepancies, more research on MHT by genotype interactions is needed.

The current work represents the most comprehensive study of detailed MHT data, APOE ε4 genotype, and several brain measures in a large population-based cohort to date. Overall, our findings do not unequivocally support general neuroprotective effects of MHT, nor do they indicate severe adverse effects of MHT use on the female brain. The results suggest subtle yet complex relationships between MHT's and brain health, highlighting the necessity for a personalized approach to MHT use. Importantly, our analyses provide a broad view of population-based associations and are not designed to guide individual-level decisions regarding the benefits versus risks of MHT use. Furthermore, several study limitations should be acknowledged. The presented data does not enable causal inference, and observational studies are subject to different sources of heterogeneity such as switching between MHT regimes (e.g. due to side effects or availability) and variable MHT formulation and dose. In addition, utilizing prescription registry data comes with its own set of challenges. In the UK, a national system for collecting or sharing primary care data is currently missing, and the availability, completeness, and level of detail in the data might vary between systems, suppliers, and over time. The primary care data used in the current study was drawn from an interim release, including data on approximately 231,000 participants. Hence, our analyses conducted on these participants' data and the observed results might not generalize to the entire cohort. Furthermore, although we extracted prescription MHT data, medications that were prescribed were not necessarily dispensed or used. Moving forward, research utilizing unified prescription registry data is needed to overcome some of these hurdles. In addition, previous studies highlight that UK Biobank participants are considered healthier than the general population based on several lifestyle and health-related factors (*Golomb et al., 2012*; *Bradley and Nichols, 2022*). This healthy volunteer bias increases with age, likely resulting in a disproportionate number of healthier older adults. Together with the imbalance in age distributions across groups, this might explain the less apparent brain aging in the older MHT user groups. We have previously highlighted that age is negatively associated with the number of APOE ε4 carriers in the UK Biobank (*Lange et al., 2020*), which is indicative of survivor bias. In addition to these inherent biases in aging cohorts, the ethnic background of the sample is homogeneous (>96% white), further reducing the generalizability of the results. Lastly, although the UK Biobank has a wealth of female-specific variables, the acquired data relies on self-reports, which might not be reliable and data recording does not always align with best practice standards. For example, menopausal status in the UK Biobank is recorded based on whether the menstrual period has generally stopped, not whether it has been absent for at least 12 mo, in line with the STRAW criteria (*Harlow et al., 2012*).

In conclusion, our findings suggest that associations between MHT use and female brain health might vary depending on duration of use and past surgical history. Although the effect sizes were generally modest, future longitudinal studies and RCTs, particularly focused on the perimenopausal transition window, are warranted to fully understand how MHT use influences female brain health. Importantly, considering risks and benefits, decisions regarding MHT use should be made within the clinical context unique to each individual.

## Data availability statement

This research has been conducted using the UKB under Application 27412. The data that support the findings of this study are available through the UK Biobank application procedure (https://www.ukbiobank.ac.uk/enable-your-research/register). Code to extract prescription MHT data from primary care records (https://github.com/MelisAnaturk/dementia_risk_score/tree/main/scripts) and to run brain age prediction (https://github.com/dmlc/xgboost) is available on Github.

## Acknowledgements

We thank Dr. Melis Anatürk for sharing her prescription registry data extraction script, Brian F O'Donnell for assistance with prescription data extraction, Dr. Dennis van der Meer for the extraction of the APOE e4 genotype, Prof. Tobias Kaufman, and Dr. Ivan I Maximov for establishing the MRI preprocessing infrastructure used for brain age prediction, and Dr. Caitlin Taylor for her insights on female health at the beginning of the project. This research has been conducted using the UKB under Application 27412. The UKB has received ethics approval from the National Health Service National Research Ethics Service (ref 11/NW/0382). The analyses were performed on the Service for Sensitive Data (TSD) platform, owned by the University of Oslo, operated, and developed by the TSD service group at the University of Oslo IT-Department (USIT). Computations were also performed using resources provided by UNINETT Sigma2 - the National Infrastructure for High-Performance Computing and Data Storage in Norway.

## Additional information

### Funding

| Funder | Grant reference number | Author |
| --- | --- | --- |
| Helse Sør-Øst RHF | 2023037 | Claudia Barth |
| European Research Council | 10.3030/802998 | Lars T Westlye |
| Helse Sør-Øst RHF | 2018076 | Lars T Westlye |
| Schweizerischer Nationalfonds zur Förderung der Wissenschaftlichen Forschung | PZ00P3_193658 | Ann-Marie G de Lange |
| Norges Forskningsråd | 223273 | Lars T Westlye |
| Canadian Institutes of Health Research | PJT-173554 | Liisa AM Galea |
| National Institutes of Health | AG063843 | Emily G Jacobs |
| Helse Sør-Øst RHF | 2022103 | Claudia Barth |
| Helse Sør-Øst RHF | 2019101 | Lars T Westlye |
| Norges Forskningsråd | 249795 | Lars T Westlye |
| Norges Forskningsråd | 273345 | Lars T Westlye |
| Norges Forskningsråd | 298646 | Lars T Westlye |
| Norges Forskningsråd | 300768 | Lars T Westlye |
| The Ann S. Bowers Women's Brain Health Initiative | | Emily G Jacobs |

The funders had no role in study design, data collection and interpretation, or the decision to submit the work for publication.

### Author contributions

Claudia Barth, Conceptualization, Data curation, Software, Formal analysis, Investigation, Visualization, Methodology, Writing – original draft, Project administration, Writing – review and editing; Liisa AM Galea, Emily G Jacobs, Bonnie H Lee, Conceptualization, Writing – review and editing; Lars T Westlye, Conceptualization, Methodology, Writing – review and editing; Ann-Marie G de Lange, Conceptualization, Resources, Software, Methodology, Writing – original draft, Writing – review and editing

## Author ORCIDs
Claudia Barth  https://orcid.org/0000-0001-6544-0945
Emily G Jacobs  http://orcid.org/0000-0003-0001-5096
Lars T Westlye  http://orcid.org/0000-0001-8644-956X

## Ethics

This research has been conducted using the UKB under Application 27412. The UKB has received ethics approval from the National Health Service National Research Ethics Service (ref 11/NW/0382). The analyses were performed on the Service for Sensitive Data (TSD) platform, owned by the University of Oslo, operated, and developed by the TSD service group at the University of Oslo IT-Department (USIT). Computations were also performed using resources provided by UNINETT Sigma2 - the National Infrastructure for High Performance Computing and Data Storage in Norway.

Reviewer #1 (Public review): https://doi.org/10.7554/eLife.99538.3.sa1
Reviewer #2 (Public review): https://doi.org/10.7554/eLife.99538.3.sa2
Author response https://doi.org/10.7554/eLife.99538.3.sa3

---

# Additional files

## Supplementary files

Supplementary file 1. Assessment of menopausal hormone therapy (MHT)-related variables in the UK Biobank (UKB).

Supplementary file 2. Lifestyle factors, constituting the lifestyle score, in menopausal hormone therapy (MHT) never-, current, and past- users in the whole sample.

Supplementary file 3. Age prediction accuracy for the global gray and white matter models.

Supplementary file 4. Associations between menopausal hormone therapy (MHT)-related variables and brain measures in the whole sample.

Supplementary file 5. Associations between menopausal hormone therapy (MHT)-related variables and brain measures in the prescription MHT sample.

Supplementary file 6. Associations between apolipoprotein ε type 4 (APOE ε4) genotype and brain measures in the whole sample.

Supplementary file 7. Interactions between apolipoprotein ε type 4 (APOE ε4) genotype and menopausal hormone therapy (MHT)-related variables on brain measures in the entire sample.

Supplementary file 8. Interactions between apolipoprotein ε type 4 (APOE ε4) genotype and menopausal hormone therapy (MHT)-related variables on brain measures in the prescription sample.

Supplementary file 9. Associations between menopausal hormone therapy (MHT)-related variables and brain measures in the whole sample, excluding participants with ICD-10 diagnosis known to impact the brain.

Supplementary file 10. Associations between menopausal hormone therapy (MHT)-related variables and brain measures in the prescription MHT sample, excluding participants with ICD-10 diagnosis known to impact the brain.

Supplementary file 11. Associations between menopausal hormone therapy (MHT)-related variables and brain measures in the whole sample, also adjusting for age (*Ding et al., 2013*).

Supplementary file 12. Associations between menopausal hormone therapy (MHT)-related variables and brain measures in the whole sample, after removal of extreme values.

Supplementary file 13. Detected extreme values of continuous menopausal hormone therapy (MHT)-related variables using the median absolute deviation method.

Supplementary file 14. Associations between menopausal hormone therapy (MHT)-related variables and brain measures in the prescription MHT sample, adjusting for additional covariates.

Supplementary file 15. Associations between menopausal hormone therapy (MHT)-related variables and brain measures in the prescription MHT sample, with age, education, and menopause-status matched never-users.

MDAR checklist

## Data availability

This research has been conducted using the UKB under Application 27412. The data that support the findings of this study are available through the UK Biobank application procedure (https://www.ukbiobank.ac.uk/enable-your-research/register). Code to extract prescription MHT data from primary care records (https://github.com/MelisAnaturk/dementia_risk_score/tree/main/scripts; *Patel and Anaturk, 2023*) and to run brain age prediction (https://github.com/dmlc/xgboost; *XGBoost, 2025*) is available on Github.

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

## Appendix

### Note 1| White matter bain age estimation

Metric from four diffusion models were utilized to predict white matter brain age, namely diffusion tensor imaging (DTI) (*Basser et al., 1994*), diffusion kurtosis imaging (DKI) (*Jensen et al., 2005*), white matter tract integrity (WMTI) (*Fieremans et al., 2011*), and spherical mean technique (SMT) (*Kaden et al., 2016a*; *Kaden et al., 2016b*). The DTI metrics included mean diffusivity (MD), fractional anisotropy (FA), axial diffusivity (AD), and radial diffusivity (RD) (*Basser et al., 1994*). The DKI metrics included mean kurtosis (MK), axial kurtosis (AK), and radial kurtosis (RK) (*Jensen et al., 2005*). WMTI metrics included axonal water fraction (AWF), extra-axonal axial diffusivity (axEAD), and extra-axonal radial diffusivity (radEAD) (*Fieremans et al., 2011*). SMT metrics included intra-neurite volume fraction (INVF), extra-neurite mean diffusivity (exMD), and extra-neurite radial diffusivity (exRD) (*Kaden et al., 2016a*). For each diffusion model metric, WM features were extracted based on John Hopkins University (JHU) atlases for white matter tracts (with 0 thresholding) (*Mori et al., 2005*, including mean values and regional measures for 12 tracts) *Voldsbekk et al., 2021*; *Krogsrud et al., 2016*; *Westlye et al., 2010*: anterior thalamic radiation (ATR), corticospinal tract (CST) cingulate gyrus (CG), cingulum hippocampus (CING), forceps major (FMAJ), forceps minor (FMIN), inferior fronto-occipital fasciculus (IFOF), inferior longitudinal fasciculus (ILF) superior longitudinal fasciculus (SLF), uncinate fasciculus (UF), superior longitudinal fasciculus temporal (SLFT), and corpus callosum (CC).

### Note 2| ICD-10 diagnosis

We excluded participants with diagnosed brain disorders (n=1739) from the brain age prediction and in sensitivity analyses, established based on ICD-10 criteria, including field F ('Mental and behavioral disorders') with F00-F03 for Alzheimer's disease and dementia and F06.7 ('Mild cognitive disorder'), field G ('Diseases of the nervous system') with inflammatory and neurodegenerative diseases (except G55-59; 'Nerve, nerve root and plexus disorders') and field I ('Diseases of the circulatory system') with I64 for stroke. An overview of the diagnoses is provided in the UK Biobank online resources (https://biobank.ndph.ox.ac.uk/showcase/field.cgi?id=41270), and the diagnostic criteria are listed in the ICD10 diagnostic manual (https://www.who.int/classifications/icd/icdonlineversions).

### Note 3| Lifestyle score

The lifestyle score was calculated based on sleep duration, time spent watching television, current and past smoking status, alcohol consumption frequency, physical activity level (number of days per week of moderate/vigorous activity for at least 10 min), intake of fruits and vegetables, and intake of oily fish, beef, lamb/mutton, pork, and processed meat (for details see *Foster et al., 2018*). Each unhealthy lifestyle factor was scored with 1 point (e.g. smoking), and participants points were summed to generate an unweighted score (from 0 to 9): the higher the lifestyle score, the unhealthier the participant's lifestyle.

A comparison of the lifestyle factors contained in the lifestyle score by MHT user status is presented in *Supplementary file 2*. In summary, we found that current MHT were more often smokers than never-users, had a higher alcohol intake than never- and past MHT users, reported the lowest fruit and vegetable intake relative to never-users and past MHT users, and stated lower moderate activity levels relative to past MHT users. Past MHT users reported higher alcohol intake than never-users, spend more time watching TV relative to never- and current users, consumed more beef, pork, lamb/mutton, and processed meat than never-users, and reported lower vigorous activity levels relative to never-users. However, oily fish intake and fruit and vegetable intake was higher among past MHT users relative to never-and current users. Self-reported sleep duration did not differ between MHT user groups.

### Note 4| Detailed results in the prescription sample not surviving correction for multiple comparison

In the prescription sample, we observed several interactions between APOE ε4 status and MHT variables on MRI measures before adjusting for multiple comparisons. Mixed bioidentical & synthetic estrogens + progestin use also showed an interaction with APOE ε4 genotype on right hippocampus volume ($\beta$=−0.292, p=0.007, $p_{FDR}$ = 0.449). Among synthetic estrogens + progestin users, CEE & norgestrel use showed an interaction with APOE ε4 genotype on GM BAG ($\beta$=−0.513, p=0.042, $p_{FDR}$ = 0.561) and WMH volumes ($\beta$=−0.499, p=0.022, $p_{FDR}$ = 0.449). Tibolone use ($\beta$=0.881, p=0.044,

$p_{FDR}$ = 0.561) and estradiol hemihydrate & norethisterone acetate use ($\beta$=−0.320, p=0.022, $p_{FDR}$ = 0.449) both showed an interaction with APOE ε4 genotype on left and right hippocampus volume, respectively. First generation progestin use ($\beta$=−0.231, p=0.018, $p_{FDR}$ = 0.449) and mean progestin dosage ($\beta$=0.512, p=0.010, $p_{FDR}$ = 0.449) also interacted with APOE ε4 genotype on right hippocampus volume and WMH volume, respectively. Concerning mode of administration, injection showed an interaction with APOE ε4 genotype on left hippocampus volume ($\beta$=−0.703, p=0.022, $p_{FDR}$ = 0.449).

