## [Editor Report · eLife Assessment]

This observational study from the UK Biobank provides an **important** investigation into the associations between menopausal hormone therapy and brain health in a large, population-based cohort of females in the UK. A **convincing** model of brain aging using an open source algorithm is used. While some modest adverse brain health characteristics were associated with current mHT use and older age at last use, the findings do not support a general neuroprotective effect of mHT nor severe adverse effects on the female brain. This work addresses a topic that is of grave importance since menopausal hormone therapy and its effect on the brain should be better understood in order to provide individualized effective medical support to women going through menopause.

---

## [Referee Report · Reviewer #1 (Public review)]

Summary:

This study takes a detailed approach to understand the effect of menopausal hormone therapy (MHT) in brain aging of females. Neuroimaging data from the UK Biobank is used to explore brain aging and shows an unexpected effect of current MHT use and poorer brain health outcomes relative to never users. There is considerable debate about the benefits of MHT and estrogens in particular for brain health, and this analysis illustrates thta the effects are certainly not straight forward and require greater considerations.

Strengths:

(1) The detailed approach to obtain important information about MHT use from primary care records. Prior studies have suggested that factors such as estrogen/progestin type, route of administration, duration, and timing of use relative to menopause onset can contribute to whether MHT benefits brain health.

(2) Consideration of type of menopause (spontaneous, or surgical) in the analysis, as well as sensitivity diagnoses to rule out the effect being driven by those with clinical conditions

(3) The incorporation of the brain age estimate along with hippocampal volume to address brain health

(4) The complex data are also well explained and interpretations are reasonable.

(5) Limitations of the UKbiobank data are acknowledged

Weaknesses:

These have since been addressed by the authors in the revision.

---

## [Referee Report · Reviewer #2 (Public review)]

Summary:

In this observational study, Barth et al. investigated the association between menopausal hormone therapy and brain health in middle- to older-aged women from the UK Biobank. The study evaluated detailed MHT data (never, current, or past user), duration of mHT use (age first/last used), history of hysterectomy with or without bilateral oophorectomy, APOEE4 genotype, and brain characteristics in a large, population-based sample. The researchers found that current mHT use (compared to never-users), but not past use, was associated with a modest increase in gray and white matter brain age gap (GM and WM BAG) and decrease in hippocampal volumes. No significant association was found between the age of mHT initiation and brain measures among mHT users. Longer duration of use and older age at last MHT use post-menopause were associated with higher GM and WM BAG, larger WMH volumes, and smaller hippocampal volumes. In a sub-sample, after adjusting for multiple comparisons, no significant associations were found between detailed mHT variables (formulations, route of administration, dosage) and brain measures. The association between mHT variables and brain measures was not influenced by APOEE4 allele carrier status. Women with a history of hysterectomy with or without bilateral oophorectomy had lower GM BAG compared to those without such history. Overall, these observational data suggest that the association between mHT use and brain health in women may vary depending on the duration of use and surgical history.

Strengths:

The study has several strengths, including a large, population-based sample of women in the UK, and comprehensive details of demographic variables such as menopausal status, history of oophorectomy/hysterectomy, genetic risk factors for Alzheimer's disease (APOE ε4 status), age at mHT initiation, age at last use, duration of mHT, and brain imaging data (hippocampus and WMH volume).

In a sub-sample, the study accessed detailed mHT prescription data (formulations, route of administration, dosage, duration), allowing the researchers to study how these variables were associated with brain health outcomes. This level of detail is generally missing in observational studies investigating the association of mHT use with brain health.

Weaknesses:

While the study has many strengths, it also has some weaknesses. These weaknesses were properly discussed throughout the article. The manuscript has indicated that the need of mHT use which might be associated with these symptoms may be indicators of preexisting neurological changes, potentially reflecting worse brain health scores, including higher BAG and lower hippocampal volume and/or higher WMH. The authors noted that the UK Biobank lacks detailed information on menopausal symptoms and perimenopausal staging, limiting the study's ability to understand how these variables influence outcomes. The authors also highlighted that these results don't reflect causal relationships. The authors caution that these findings should not guide individual-level decisions regarding the benefits versus risks of mHT use. However, the study raises new questions that should be addressed by randomized clinical trials to investigate the varying effects of MHT on brain health and dementia risk.

---

## [Author Response]

The following is the authors’ response to the original reviews

**Reviewer #1 (Public review):**
Summary:This study takes a detailed approach to understanding the effect of menopausal hormone therapy (MHT) in the brain aging of females. Neuroimaging data from the UK Biobank is used to explore brain aging and shows an unexpected effect of current MHT use and poorer brain health outcomes relative to never users. There is considerable debate about the benefits of MHT and estrogens in particular for brain health, and this analysis illustrates that the effects are certainly not straightforward and require greater consideration.Strengths:(1) The detailed approach to obtaining important information about MHT use from primary care records. Prior studies have suggested that factors such as estrogen/progestin type, route of administration, duration, and timing of use relative to menopause onset can contribute to whether MHT benefits brain health.(2) Consideration of type of menopause (spontaneous, or surgical) in the analysis, as well as sensitivity diagnoses to rule out the effect being driven by those with clinical conditions.(3) The incorporation of the brain age estimate along with hippocampal volume to address brain health.(4) The complex data are also well explained and interpretations are reasonable.(5) Limitations of the UK Biobank data are acknowledged

We thank the reviewer for their time and the positive evaluation of our manuscript.

Weaknesses:(1) Lifestyle factors are listed and the authors acknowledge group differences (at least between current users and never users of MHT). I was not able to find these analyses showing these differences.

We highlighted and tested for group differences in lifestyle scores, and the results are shown in Table 1-3, column p-value. As highlighted in the method section (page 9): “The lifestyle score was calculated using a published formula (69), and included data on sleep, physical activity, nutrition, smoking, and alcohol consumption (see supplementary Note 3, Table S2)”. In line with reviewer 1 suggestion to the authors, we now included an additional table testing for group differences in the specific lifestyle factors constituting the lifestyle score in the supplementary materials (Table S2). Please find a more detailed response below (Recommendations for the authors, Response to Comment 1).

(2) The distribution of women who were not menopausal was unequal across groups, and while the authors acknowledge this, one wonders to what extent this explains the observed findings.

We agree with the reviewer that the unequal distribution of women across groups can influence the observed findings. We have made minor edits to highlight this important topic more explicitly in the discussion:

Discussion (page 21): “Current MHT users were significantly younger than past- and never-users, and around 67 % were menopausal relative to over 80% in the past- and never-user groups. The unequal distribution of age and menopausal status across groups may have influenced the observed findings. For instance, a larger proportion of the current users might be in the perimenopausal phase, which is often associated with debilitating neurological and vasomotor symptoms (1). MHT is commonly prescribed to minimize such symptoms. Although MHT initiation during perimenopause has been associated with improved memory and hippocampal function, as well as lower AD risk later in life (15), the need for MHT might in itself be an indicator of neurological changes (71); here potentially reflected in higher BAG and lower hippocampal volumes. After the transition to menopause, symptoms might subside and some perimenopausal brain changes might revert or stabilize in the postmenopausal phase 5. Although the UK Biobank lacks detailed information on menopausal symptoms and perimenopausal staging, our results might be capturing subtle disturbances during perimenopause that later stabilize. This could explain why the largely postmenopausal groups of past MHT users and never-users present with lower GM and WM BAG than the current user group. Considering the critical window hypothesis emphasizing perimenopause as a key phase for MHT action (29,43), future longitudinal studies are crucial to clarify the interplay between neurological changes and MHT use across the menopause transition.”

Discussion (page 25): “In addition, previous studies highlight that UK Biobank participants are considered healthier than the general population based on several lifestyle and health-related factors (89, 90). This healthy volunteer bias increases with age, likely resulting in a disproportionate number of healthier older adults. Together with the imbalance in age distributions across groups, this might explain the less apparent brain aging in the older MHT user groups. We have previously highlighted that age is negatively associated with the number of APOE ε4 carriers in the UK Biobank (21), which is indicative of survivor bias.”

(3) While the interpretations are reasonable, and relevant theories (healthy cell & critical window) are mentioned, the discussion is missing a more zoomed-out perspective of the findings. While I appreciate wanting to limit speculation, the reader is left having to synthesize a lot of complex details on their own. A particularly difficult finding to reconcile is under what conditions these women benefit from MHT and when do they not (and why that may be).

We thank the reviewer for this comment. As the presented data is cross-sectional and does not enable causal inference, we have refrained from a more zoomed-out interpretation of the results to avoid undue speculations. However, where applicable, we have discussed our findings in a broader context such as the effects of MHT use on the brain across the menopausal transition (discussion page 21) and the effects of MHT use on the brain in the presence and absence of bilateral oophorectomy and/or hysterectomy (discussion page 25).

To best inform the reader about the scope of our paper, we would like to highlight the following sentences in our discussion (page 24):

“The current work represents the most comprehensive study of detailed MHT data, APOE ε4 genotype, and several brain measures in a large population-based cohort to date. Overall, our findings do not unequivocally support general neuroprotective effects of MHT, nor do they indicate severe adverse effects of MHT use on the female brain. The results suggest subtle yet complex relationships between MHT’s and brain health, highlighting the necessity for a personalized approach to MHT use. Importantly, our analyses provide a broad view of population-based associations and are not designed to guide individual-level decisions regarding the benefits versus risks of MHT use.”

And the conclusion (page 25): “In conclusion, our findings suggest that associations between MHT use and female brain health might vary depending on duration of use and past surgical history. Although the effect sizes were generally modest, future longitudinal studies and RCTs, particularly focused on the perimenopausal transition window, are warranted to fully understand how MHT use influences female brain health. Importantly, considering risks and benefits, decisions regarding MHT use should be made within the clinical context unique to each individual.”

**Reviewer #1 (Recommendations for the authors):**
Can the authors provide:(1) More information about which aspects of lifestyle factors were different between the groups, and how these factors may have contributed to the observed findings (if possible, without burying this information in the supplemental)?

We thank the reviewer for this suggestion. We now added a table comparing lifestyle factors contained in the lifestyle score by MHT user status using t-tests (continuous variables) or χ2 tests (see Table S2). The results are referred to in the main manuscript result section under “Sample characteristics”, and the table (Table S2) is provided in the supplements not to overburden the main text, in line with input from reviewer 3.

We updated the main text to refer to Table S2 and updated the supplementary Note 3 (page 2-3) to include the results of the comparison of the lifestyle factors contained in the lifestyle score by MHT user status.

Methods, page 9:“The lifestyle score was calculated using a published formula (69), and included data on sleep, physical activity, nutrition, smoking, and alcohol consumption (see supplementary Note 3, Table S2).”

Results, page 13: “Sample demographics including lifestyle score, stratified by MHT user group, surgical history among MHT users, and estrogen only MHT or combined MHT use, are summarized in Table 1, 2 and 3, respectively. MHT user group differences for each lifestyle factor contained in the lifestyle score are shown in Table S2.”

“Note 3| Lifestyle Score

The lifestyle score was calculated based on sleep duration, time spent watching television, current and past smoking status, alcohol consumption frequency, physical activity level (number of days per week of moderate/vigorous activity for at least 10 minutes), intake of fruits and vegetables, and intake of oily fish, beef, lamb/mutton, pork and processed meat (for details see (10)). Each unhealthy lifestyle factor was scored with 1 point (e.g., smoking), and participants points were summed to generate an unweighted score (from 0-9): the higher the lifestyle score, the unhealthier the participant’s lifestyle.

A comparison of the lifestyle factors contained in the lifestyle score by MHT user status is presented in Table S2. In summary, we found that current MHT were more often smokers than never-users, had a higher alcohol intake than never- and past MHT users, reported the lowest fruit and vegetable intake relative to never-users and past MHT users, and stated lower moderate activity levels relative to past MHT users. Past MHT users reported higher alcohol intake than never-users, spend more time watching TV relative to never- and current-users, consumed more beef, pork, lamb/mutton, and processed meat than never-users, and reported lower vigorous activity levels relative to never-users. However, oily fish intake and fruit and vegetable intake was higher among past MHT users relative to never-and current-users. Self-reported sleep duration did not differ between MHT user groups.”

(2) A greater description of the 2 main theories of MHT effects on the brain (healthy cell vs critical window). Can the authors also provide a more thorough explanation for how the findings fit with these theories.

We thank the reviewer for this comment. We have described our findings in the context of the critical window hypothesis (discussion, page 21, paragraph 2), the healthy cell bias hypothesis (discussion, page 22, paragraph 3), and healthy user bias hypothesis (discussion, page 22, paragraph 4). We refrained from a more thorough explanation to avoid undue speculations.

(3) Reflect more on what the findings may indicate as to who benefits from MHT, and why. There are some references that the authors may want to add, particularly related to recent findings from premenopausal bilateral oophortectomies that also speak to when (and for whom) MHT use might benefit.

We thank the reviewer for this feedback. We have included additional references in the revised manuscript as follows:

Discussion, page 23: “It is also possible that the timing between MHT use and surgery is more tightly controlled and therefore more beneficial for brain aging (43). For instance, studies suggest that MHT may mitigate the potential long-term adverse effects of bilateral oophorectomy before natural menopause on bone mineral density as well as cardiovascular, cognitive and mental health (79-81). In addition, a 2024 UK Biobank study found that ever used MHT was associated with decreased odds of Alzheimer’s disease in women with bilateral oophorectomy (82).”

(79) Blumel JE, Arteaga E, Vallejo MS, et al. Association of bilateral oophorectomy and menopause hormone therapy with mild cognitive impairment: the REDLINC X study. Climacteric 2022;25:195-202.

(80) Kaunitz AM, Kapoor E, Faubion S. Treatment of Women After Bilateral Salpingo-oophorectomy Performed Prior to Natural Menopause. JAMA 2021;326:1429-1430.

(81) Stuursma A, Lanjouw L, Idema DL, de Bock GH, Mourits MJE. Surgical Menopause and Bilateral Oophorectomy: Effect of Estrogen-Progesterone and Testosterone Replacement Therapy on Psychological Well-being and Sexual Functioning; A Systematic Literature Review. J Sex Med 2022;19:1778-1789.

(82) Calvo N, McFall GP, Ramana S, et al. Associated risk and resilience factors of Alzheimer's disease in women with early bilateral oophorectomy: Data from the UK Biobank. J Alzheimers Dis 2024;102:119-128.

**Reviewer #2 (Public review):**
Summary:In this observational study, Barth et al. investigated the association between menopausal hormone therapy and brain health in middle- to older-aged women from the UK Biobank. The study evaluated detailed MHT data (never, current, or past user), duration of mHT use (age first/last used), history of hysterectomy with or without bilateral oophorectomy, APOEE4 genotype, and brain characteristics in a large, population-based sample. The researchers found that current mHT use (compared to never-users), but not past use, was associated with a modest increase in gray and white matter brain age gap (GM and WM BAG) and a decrease in hippocampal volumes. No significant association was found between the age of mHT initiation and brain measures among mHT users. Longer duration of use and older age at last MHT use post-menopause were associated with higher GM and WM BAG, larger WMH volumes, and smaller hippocampal volumes. In a sub-sample, after adjusting for multiple comparisons, no significant associations were found between detailed mHT variables (formulations, route of administration, dosage) and brain measures. The association between mHT variables and brain measures was not influenced by APOEE4 allele carrier status. Women with a history of hysterectomy with or without bilateral oophorectomy had lower GM BAG compared to those without such a history. Overall, these observational data suggest that the association between mHT use and brain health in women may vary depending on the duration of use and surgical history.Strengths:(1) The study has several strengths, including a large, population-based sample of women in the UK, and comprehensive details of demographic variables such as menopausal status, history of oophorectomy/hysterectomy, genetic risk factors for Alzheimer's disease (APOE ε4 status), age at mHT initiation, age at last use, duration of mHT, and brain imaging data (hippocampus and WMH volume).(2) In a sub-sample, the study accessed detailed mHT prescription data (formulations, route of administration, dosage, duration), allowing the researchers to study how these variables were associated with brain health outcomes. This level of detail is generally missing in observational studies investigating the association of mHT use with brain health.

We thank the reviewer for their time and the positive evaluation of our manuscript.

Weaknesses:(1) While the study has many strengths, it also has some weaknesses. As highlighted in an editorial by Kantarci & Manson (2023), women with symptoms such as subjective cognitive problems, sleep disturbances, and elevated vasomotor symptoms combined with sleep disturbances tend to seek mHT more frequently than those without these symptoms. The authors of this study have also indicated that the need of mHT use which might be associated with these symptoms may be indicators of preexisting neurological changes, potentially reflecting worse brain health scores, including higher BAG and lower hippocampal volume and/or higher WMH. However, among current users, how many of these women have these symptoms could not be reported in the study. Women with these vasomotor symptoms who are using mHT are more likely to stay longer in the healthcare system compared with those without these symptoms and no MHT use history. The authors noted that the UK Biobank lacks detailed information on menopausal symptoms and perimenopausal staging, limiting the study's ability to understand how these variables influence outcomes.

We thank the reviewer for the succint synopsis of the limitations highlighted in discussion, page 21. We have now added the mentioned reference, 2023 editoral by Kantarci & Manson, to the discussion as well (see reference 71).

Discussion (page 21): “Current MHT users were significantly younger than past- and never-users, and around 67 % were menopausal relative to over 80% in the past- and never-user groups. The unequal distribution of age and menopausal status across groups may have influenced the observed findings. For instance, a larger proportion of the current users might be in the perimenopausal phase, which is often associated with debilitating neurological and vasomotor symptoms (1). MHT is commonly prescribed to minimize such symptoms. Although MHT initiation during perimenopause has been associated with improved memory and hippocampal function, as well as lower AD risk later in life (15), the need for MHT might in itself be an indicator of neurological changes (71); here potentially reflected in higher BAG and lower hippocampal volumes. After the transition to menopause, symptoms might subside and some perimenopausal brain changes might revert or stabilize in the postmenopausal phase 5. Although the UK Biobank lacks detailed information on menopausal symptoms and perimenopausal staging, our results might be capturing subtle disturbances during perimenopause that later stabilize. This could explain why the largely postmenopausal groups of past MHT users and never-users present with lower GM and WM BAG than the current user group. Considering the critical window hypothesis emphasizing perimenopause as a key phase for MHT action (29,43), future longitudinal studies are crucial to clarify the interplay between neurological changes and MHT use across the menopause transition.”

(2) Earlier observational studies have reported conflicting results regarding the association between mHT use and the risk of dementia and brain health. Contrary to some observational studies, three randomized trials (WHI, KEEPS, ELITE) (Espeland et al 2013, Gleason et al 2015; Henderson et al 2016) demonstrated neither beneficial nor harmful effects of mHT (with varying doses and formulations) when initiated closer to menopause (<5 years). While strong efforts were made to run proper statistical analyses to investigate the association between mHT use and brain health, these results reflect mainly associations, but not causal relationships as also stated by the authors.

We thank the reviewer for pointing that out.

(3) Furthermore, observational studies have intrinsic limitations, such as a lack of control over switching mHT doses and formulations, a lack of laboratory measures to confirm mHT use, and reliance on self-reported data, which may not always be reliable. The authors caution that these findings should not guide individual-level decisions regarding the benefits versus risks of mHT use. However, the study raises new questions that should be addressed by randomized clinical trials to investigate the varying effects of MHT on brain health and dementia risk.

We thank the reviewer for making our efforts in providing proper disclaimers in the discussion visible.

**Reviewer #2 (Recommendations for the authors):**
(1) The study could benefit from extending these findings by adding plasma biomarkers of AD and PET imaging markers to further study the association of mHT variables with brain health.

We agree with the reviewer that such markers would be beneficial for elucidating the association between MHT variables and brain health. Unfortunately, these markers are not readily available in the UK Biobank.

(2) The study's reliance on a predominantly white cohort limits the generalizability of the findings to more diverse populations. This homogeneity may not capture the full spectrum of responses to MHT across different ethnic and genetic backgrounds.

We fully agree with the reviewers statement and state this limitation in the discussion (page 25) as follows:

“In addition to these inherent biases in aging cohorts, the ethnic background of the sample is homogeneous (> 96% white), further reducing the generalizability of the results.”

(3) The study may benefit by editing the following information in the introduction: "In summary, WHIMS, HERS, and KEEPS mainly relied on orally administered CEE in older-aged or recently postmenopausal females." KEEPS used two routes and formulations (transdermal estradiol and oCEE, both with micronized progesterone).

We thank the reviewer for catching this oversight. We removed the sentence to avoid ambiguities and revised the sentence specifically refering to the KEEPS study as follows:

Introduction, page 3: “In contrast, administering oral CEE or transdermal estradiol plus micronized progesterone in recently postmenopausal females did not alter cognition in the Kronos Early Estrogen Prevention Study (KEEPS) (28).”

(4) The study may benefit by editing the following statement in the introduction: "oral CEE use in combination with MPA seems to increase the risk for AD regardless of timing": I would suggest revising this statement, which is based on review article 29. The statement of the adverse effect of oCEE regardless of the time of start contradicts earlier randomized clinical findings. I think it is important to make a distinction between the outcomes of randomized control trials and observational studies. The WMIHS (Shumaker et al., 2003) (randomized control trial) reported that there was an increased risk of dementia for women who were more than 10 years from the onset of menopause when the therapy was initiated in oCEE + MPA compared to placebo. Two other long-duration randomized trials tested the effect of oral oestrogen and progesterone treatment on cognitive function in women who started treatment shortly after menopause (within 3 or 6 years) did not find evidence that treatment benefits or harms cognitive function compared with placebo (Gleason et al., 2015; Henderson et al., 2016). A short-term (4 months) randomized trial Maki et al 2007 (Maki et al., 2007) (mentioned in ref 29) reported a potential negative effect of CEE/MPA on verbal memory in women who started HT shortly after menopause (within 3 years). The study did not investigate the risk of dementia, and the duration of use of HT was short-term.

We thank the reviewer for this detailed input. After checking the provided references, we rephrased the sentence as follows:

Introduction, page 4:“Although emerging evidence supports this hypothesis (30, 31), oral CEE use in combination with MPA has been found to increase the risk for memory decline regardless of timing (26, 29, 32).”

We believe this formulation is more in line with the evidence provided by Shumaker et al. 2003, Maki et al. 2007 and the other references provided in the review paper by Maki and colleagues (mentioned in ref. 29). The reviewer further refers to Gleason et al. 2015 and Henderson et al. 2016, however both RCTs use micronized progesterone, not MPA, thereby not supporting the statement.

(26) Shumaker SA, Legault C, Rapp SR, et al. Estrogen plus progestin and the incidence of dementia and mild cognitive impairment in postmenopausal women: the Women's Health Initiative Memory Study: a randomized controlled trial. JAMA 2003;289:2651-2662.

(29) Maki PM. Critical window hypothesis of hormone therapy and cognition: a scientific update on clinical studies. Menopause 2013;20:695-709.

(32) Maki PM, Gast MJ, Vieweg AJ, Burriss SW, Yaffe K. Hormone therapy in menopausal women with cognitive complaints: a randomized, double-blind trial. Neurology 2007;69:1322-1330.

**Reviewer #3 (Public review):**
In this study Barth et al. present results of detailed analyses of the relationships between menopausal hormone therapy (MHT), APOE ε4 genotype, and measures of anatomical brain age in women in the UK Biobank. While past studies have investigated the links between some of these variables (including works by the authors themselves), this new study adds more detailed MHT variables, surgical status, and additional brain aging measures. The UK biobank sample is large, but it is a population cohort and many of the MHT measures are self-reported (as the authors point out). However, the authors present a solid analysis of the available information which shows associations between MHT user status, length of MHT use, as well as surgical status with brain age. However, as the authors themselves state, the results do not unequivocally support the neuroprotective or adverse effect of MHT on the brain. I think this work strengthens the case for the need of better-designed longitudinal studies investigating the effect of MHT on the brain in the peri/post-menopausal stage.Strengths:(1) The authors addressed the statistical analyses rigorously. For example, multiple testing corrections, outlier removal, and sensitivity analysis were performed carefully. Ample background information is provided in the introduction allowing even individuals not familiar with the field to understand the motivation behind the work. The discussion section also does a great job of addressing open questions and limitations. Very detailed results of all statistical tests are provided either in the main text or in the supplementary information.

We thank the reviewer for their time and the positive evaluation of our manuscript.

Weaknesses:(1) For me, the biggest weakness was the presentation of the results. As many variables are involved and past studies have investigated several of these questions, it would have helped to better clarify the analysis and questions that are addressed by this study in particular and what sets this work apart from past studies. The information is present in the manuscript but better organization might have helped. For example, a figure depicting the key questions near the beginning of the manuscript would have been very helpful for me. The Tables also contain a lot of information but I wonder if there might be a way to capture the most relevant information more succinctly (either in Table format or in a figure) for the main text.

We thank the reviewer for this comment. We do agree that with the large number of analyses it can be hard to keep an overview. We now added a Figure summarizing the main and sensitity analyses by sample.

(2) Another concern I had was the linear models investigating the effects of these MHT variables on the brain age gap. The authors have included "age" as one of the parameters in this analysis. I wonder if adding a quadratic age factor age2 in the model might have improved the fit since many brain phenotypes tend to show quadratic brain age effects in the 40 to 80-year age range.

We thank the reviewer for this suggestion. We have rerun the main analysis in the whole sample (model 1) with age squared as an additional covariate, and compared the gray matter brain age gap model fits using the corrected Akaike Information Criterion (AIC). All models with age squared had a better model fit than models without age squared (see Author response table 1). Hence, in the revised manuscript, we added a sensitivity analysis rerunning the model 1 with age squared to account for potential non-linear effect. The results were largely consistent. The manuscript was revised as follows to reflect the added analysis:

Sensitivity analysis (Methods, Page 11): “To test whether the results were influenced by the inclusion of participants with ICD-10 diagnosis or by non-linear effects of age, the main analyses (models 1-2) were re-run excluding the sub-sample with diagnosed brain disorders (see supplementary Note 2) or adding age(2) as additional covariate, respectively.”

Sensitivity analysis (Results, Page 20): “The results were consistent after removing participants with ICD-10 diagnoses known to impact the brain (see Table S9 for model 1 analyses and Table S10 for model 2 analyses), after additionally adjusting for age(2) (see Table S11), and after removing extreme values (see Table S12 for model 1 analyses).”

**Author response table 1. sa3table1:** Gray matter brain age gap model selection based on corrected Akaike Information Criterion (AICc).

Model name	Age ^(2)	K	AICc	/_\AICc	ModelLik	AICcWt	LL
MHT never-user/user	Yes	8	12347.35	0.00	1	1	-6165.66
	No	8	43951.59	31604.24	0	0	-21967.79
MHT never-/current-/past-user	Yes	10	40976.73	0.00	1	1	-20478.36
	No	9	41194.26	217.53	0	0	-20588.12
Age at first MHT use	Yes	9	12297.05	0.00	1	1	-6139.50
	No	8	12347.35	50.30	0	0	-6165.66
Age at first MHT use rel. menopause	Yes	8	11023.87	0.00	1	1	-5503.91
	No	7	11061.39	37.53	0	0	-5523.68
Age at last MHT use	Yes	9	9908.18	0.00	1	1	-4945.06
	No	8	9937.17	28.99	0	0	-4960.56
Age at last MHT use rel. menopause	Yes	8	9144.83	0.00	1	1	-4564.39
	No	7	9160.71	15.88	0	0	-4573.34
Duration of MHT use	Yes	9	11736.31	0.00	1	1	-5859.13
	No	8	11788.00	51.69	0	0	-5885.98
Oophorectomy	Yes	9	21082.53	0.00	1	1	-10532.25
	No	8	21154.50	71.97	0	0	-10569.24
Hysterectomy	Yes	9	16161.19	0.00	1	1	-8071.58
	No	8	16220.38	59.19	0	0	-8102.18

Abbreviations and explanations of parameters: MHT = menopausal hormone therapy, K = number of estimated parameters for each model, AICc = the information criterion requested for each model, ΔAICc = the appropriate delta AIC component depending on the information criteria selectedModelLik = the relative likelihood of the model given the data, AICcWT = Akaike weights to indicate the level of support in favor of any given model being the most parsimonious among the candidate model sets, LL = log-likelihood of each model.

**Reviewer #3 (Recommendations for the authors):**
(1) Please note typo in Figures 2 and 3 legend "GM WM".

We thank the reviewer for catching this typo and we changed it to BAG GM and BAG WM for all Figures for consistency.